# Sirenian genomes illuminate the evolution of fully aquatic species within the mammalian superorder afrotheria

Ran Tian[1,12], Yaolei Zhang[2,3,4,12], Hui Kang[5,6,12], Fan Zhang[1], Zhihong Jin[1], Jiahao Wang[2,3], Peijun Zhang [5], Xuming Zhou [7,8], Janet M. Lanyon [9], Helen L. Sneath [9], Lucy Woolford[10], Guangyi Fan [2,3,4,11] ✉, Songhai Li [5,6] ✉ & Inge Seim [1,5] ✉

Sirenians of the superorder Afrotheria were the first mammals to transition from land to water and are the only herbivorous marine mammals. Here, we generated a chromosome-level dugong (*Dugong dugon*) genome. A comparison of our assembly with other afrotherian genomes reveals possible molecular adaptations to aquatic life by sirenians, including a shift in daily activity patterns (circadian clock) and tolerance to a high-iodine plant diet mediated through changes in the iodide transporter NIS (*SLC5A5*) and its co-transporters. Functional in vitro assays confirm that sirenian amino acid substitutions alter the properties of the circadian clock protein PER2 and NIS. Sirenians show evidence of convergent regression of integumentary system (skin and its appendages) genes with cetaceans. Our analysis also uncovers gene losses that may be maladaptive in a modern environment, including a candidate gene (*KCNK18*) for sirenian cold stress syndrome likely lost during their evolutionary shift in daily activity patterns. Genomes from nine Australian locations and the functionally extinct Okinawan population confirm and date a genetic break ~10.7 thousand years ago on the Australian east coast and provide evidence of an associated ecotype, and highlight the need for whole-genome resequencing data from dugong populations worldwide for conservation and genetic management.

The terrestrial ancestors of the marine mammal groups Sirenia, Cetacea, and Pinnipedia independently transitioned from land to water[1]. The ostensibly first to leave land, sirenians, emerged around 60 million years ago within the afrotherian herbivorous clade Paenungulata, 10 and 30 million years before the emergence of cetaceans and pinnipeds[2]. Afrotherian mammals were isolated from other mammals until ~60 Mya when non-afrotherian mammals (ungulates from ~25 Mya) began to enter the African continent from Eurasia and displaced many local species[3,4]. This geographic isolation allowed the independent evolution of terrestrial mammals to an aquatic habitat in Africa (sirenians),

paralleling the evolution of fully aquatic cetaceans from ungulates elsewhere.

Dozens of sirenian species have existed in the past, but unlike cetaceans (about 90 extant species) and pinnipeds (about 30 extant species), sirenians are today far less diverse (Fig. S1a). There are four extant sirenian species: the dugong (*Dugong dugon*) of the family Dugongidae (included the Steller's sea cow, *Hydrodamalis gigas*, that became extinct about 250 years ago) and manatees (family Trichechidae: the West Indian manatee, *Trichechus manatus* (includes the subspecies Florida manatee, *T. m. latirostris*, and Antillean manatee, *T. m. manatus*); the Amazonian manatee, *T. inunguis*; and the African

manatee, *T. senegalensis*)[1,5]. The dugong and manatees have been found in distinct tropical and subtropical habitats since the middle Miocene (~12.2 Mya)[2,5]. The dugong originally dispersed into the Pacific from near Florida, and today inhabits the coastlines of the Indo-Pacific oceans, while the three species of manatee occupy the Atlantic Ocean and associated rivers (Fig. S1b). Although dugongs are abundant along the tropical waters off Australia, their numbers elsewhere have dwindled in recent decades and some populations are now functionally extinct. The species is listed as *Vulnerable* globally by the International Union for Conservation of Nature (IUCN)[6,7] and is threatened by habitat loss from human activities and climate change. In this study, we explored the adaptive evolution of sirenians and dugong diversity and demography. We highlight genes that may underlie sirenian adaptations, including convergent regression of integumentary system genes with cetaceans and aquatic herbivory, reconstruct demographic histories of dugong populations, and validate and characterize a genetic break in waters off the Australian East Coast.

## Results

### An annotated dugong genome for comparative and population genomic analyses

We used single-tube long fragment read (stLFR) and high-throughput chromosome conformation capture (Hi-C) sequencing to generate a 3.06 Gb 25-chromosome genome assembly of a female dugong with 18,663 annotated protein-coding genes and 1.51 Gb (49.2%) repetitive elements (Supplementary Note 1, Fig. S2, and Tables S1 and S2). The assembly and gene set BUSCO[8] completeness scores (94.4% and 91.7%, respectively) were comparable to other afrotherian species and dugong assemblies that became available after the commencement of our assembly (Tables S3 and S4).

Although similar in appearance, the dugong and manatees are not that closely related. They share a common ancestor (crown Sirenia) 31.2 Mya (95% CI: 27.4–37.0 Mya) (Fig. S1c). Our afrotherian data set, which included the West Indian manatee and the phylogenetically closest extant terrestrial species to sirenians (elephants and hyraxes) (Supplementary Note 2 and Fig. S3), was interrogated (see "Comparative genomics analysis strategy" in Methods, Table S5, and Supplementary Data 1–8) to illuminate features present since the sirenian crown ancestor (Fig. 1) that may underlie aquatic herbivory, sirenian circadian activity patterns, and typical marine mammal features such as modified cardiovascular (Supplementary Note 3), integumental (i.e., skin and associated structures), and sensory (vision, smell, and taste) systems.

### Nutrient uptake by fully aquatic herbivores

Sirenians are the only aquatic herbivorous mammals, and we observed gene losses consistent with a diet comprising few animal products (Supplementary Note 4 and Supplementary Data 8). Nearshore marine plants and aquatic plants from rivers and swamps are a rich source of iodine, a nutrient required to synthesize thyroid hormones (Fig. 2a) essential for systemic energy metabolism, thermoregulation, and the integrity of many tissues[9–11]. Both inadequate and excess iodine uptake can result in thyroid dysfunction. Despite their high-iodine plant diet, the blood thyroid hormone levels of wild West Indian manatees are unremarkable compared to the tropical, carnivorous Amazon River dolphin (*Inia geoffrensis*)[12,13]. The previous observations support the idea that genomic changes accompanied sirenian evolution from a terrestrial to an iodine-rich aquatic plant diet. In agreement, nearly all genes of the thyroid hormone pathway harbor sirenian-specific amino acid substitutions (Fig. 2b and Supplementary Data 5). These include *TSHR* (thyroid-stimulating hormone receptor), *DUXO2* (dual oxidase 2), *DUOXA2* (dual oxidase maturation factor 2), *TPO* (thyroid peroxidase), *SLC5A5* (solute carrier family 5 member 5; Fig. 2c), *KCNQ1* (potassium voltage-gated channel subfamily Q member 1), and *KCNE2* (potassium voltage-gated channel subfamily E regulatory subunit 2; Fig. 2d). We also identified positive selection of *ATP1B4* (ATPase Na⁺/K⁺ transporting family member β4), *DIO1* (iodothyronine deiodinase 1), and *DUOX2A*, and rapid evolution of *KCNE2* and the thyroid-hormone binding albumin (*ALB*) (Supplementary Data 1).

The transmembrane protein SLC5A5 (NIS) is the only known iodide transporter[14,15]. After the reduction of iodine to iodide (I⁻), iodide is transported from the bloodstream into thyroid follicular cells by NIS acting in concert with the potassium transporters KCNQ1 and KCNE2[16,17] (Fig. 2b), two proteins that also have multiple sirenian-specific amino acid substitutions. Mutations in human NIS result in congenital I⁻ deficiency disorders (IDDs)[14,18]. We identified five sirenian-specific NIS mutations. Four are in transmembrane domains (TMDs), and one is in an extracellular loop (Fig. 2c). While none of the sirenian-specific residues have been associated with IDD to date, they are close to residues conserved in mammals shown by site-directed mutagenesis to be important for NIS function. Sirenian Leu-142 flanks Tyr-144 of TMD 4[19]; sirenian Ala-445 of TMD 12 flanks Asn-441, an extracellular region residue thought to mediate NIS structure via α-helix N-capping

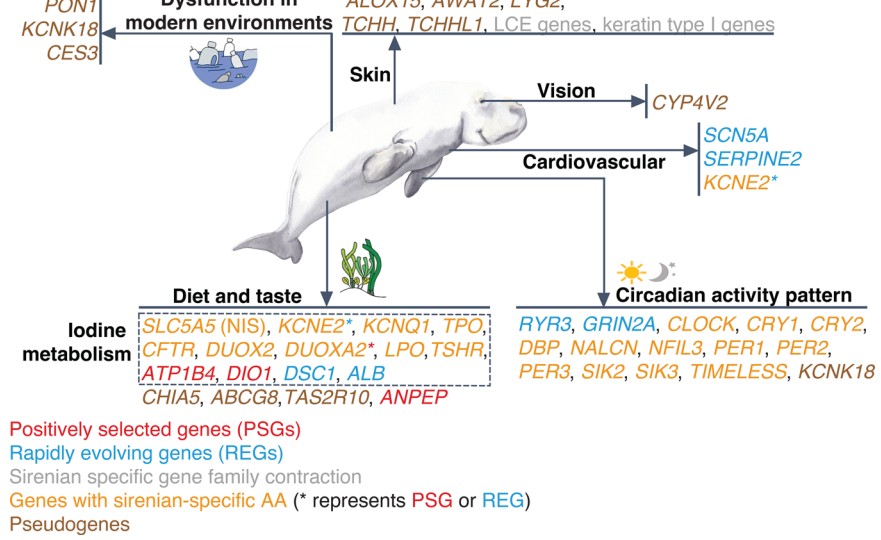

**Fig. 1 | Genes with unique evolutionary signals in sirenians.** Putative adaptive or maladaptive gene changes are shown. LCE denotes late cornified envelope. Image of dugong courtesy of Liudmila Kopecka/Shutterstock.com.

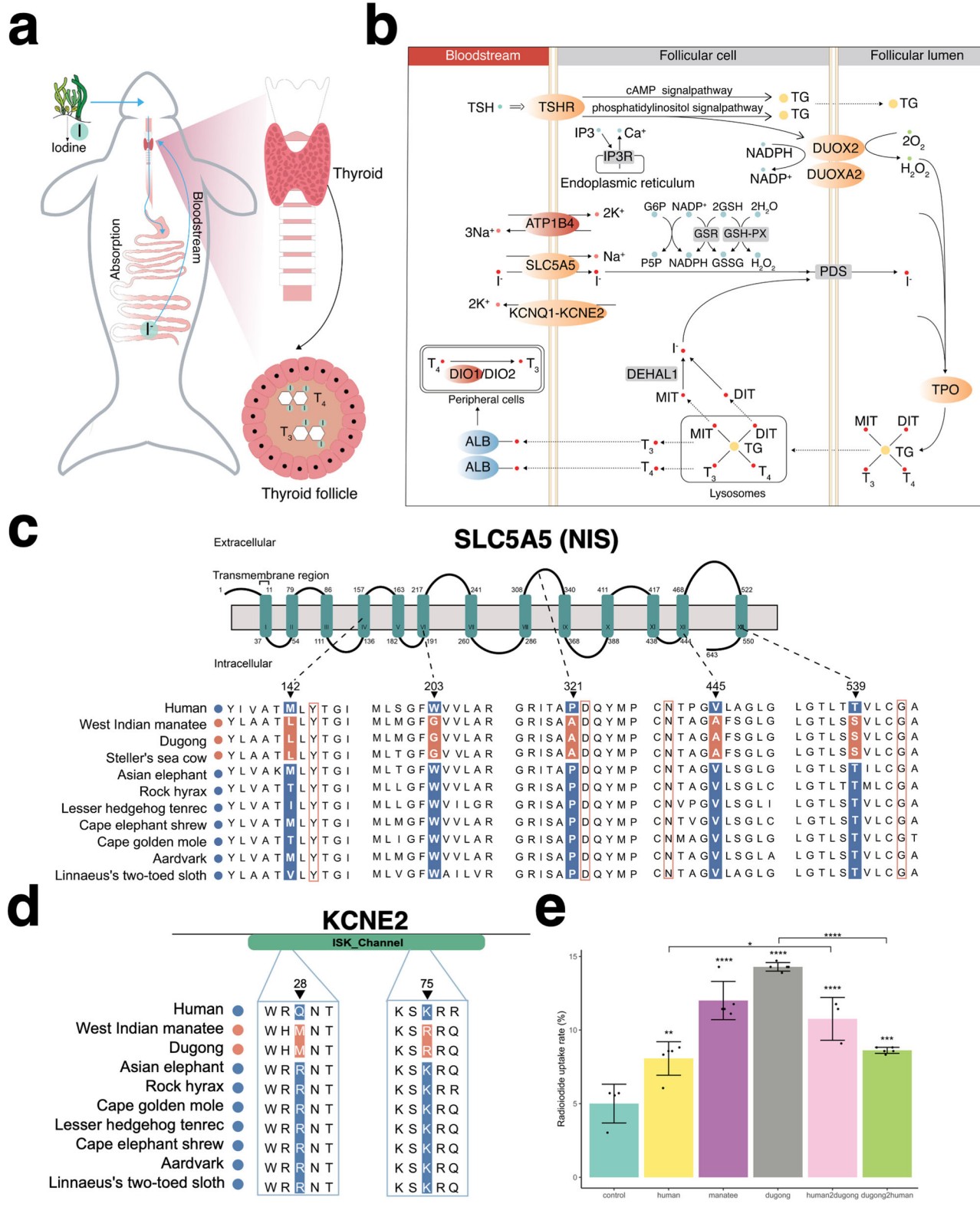

of TMD 12[20]; and sirenian Ala-321 is next to Asp-322 in the extracellular loop of TMD 8 and 9[21]. Sirenian Ser-539 flanks Gly-543 in TMD 13, a residue required for NIS cell surface targeting[22]. We carried out I⁻ uptake assays of human and sirenian NIS in HEK293T cells, revealing higher uptake of I⁻ by dugong and West Indian manatee NIS (Fig. 2e). Reciprocal site-directed mutagenesis of dugong and human NIS at the five residues further strengthens the evidence for more efficient iodide transport by the sirenian protein (Fig. 2e).

## The integumentary system

The skin of mammals consists of the epidermis (outer layer), the dermis (middle layer), and the hypodermis (deep layer) tissue. The sirenian epidermis is restructured compared to its terrestrial sister taxa. While skin appendages (hair follicles, sebaceous glands and sweat glands) are associated with the epidermis, they project deep into the dermal layer and are absent (sweat glands) or sparsely distributed (e.g., vibrissae, specialized tactile hairs scattered over the body) in

**Fig. 2 | Sirenian-specific changes in iodine uptake and organification genes.**
**a** Schematic diagram showing iodine uptake by a dugong, from seagrass ingestion to fermentation in the hindgut and reduction of iodine to iodide (I⁻), to iodide transport in the bloodstream to the thyroid. The thyroid is an endocrine gland consisting of follicles, structural units of epithelial cells enclosing a lumen (the colloid) harboring thyroglobulin (TG). TG is a precursor for thyroid hormones (indicated by $T_3$ and $T_4$) biogenesis and is also an iodide storage and carrier protein. **b** Iodide is transported from the bloodstream to thyroid follicular cells by NIS (encoded by *SLC5A5*) interacting with KCNQ1 and KCNE2. Genes with sirenian-specific amino acid substitutions in orange and genes rapidly evolving and positively selected in sirenians in blue and red, respectively. **c** Alignment showing an amino acid substitution in transmembrane domains of sirenian SLC5A5 (NIS). Sirenian-specific residues shown in red shading. Red boxes indicate residues shown to be important for NIS function[19–22]. **d** Alignment showing amino acid substitution

in the ISK_Channel domain of sirenian KCNE2. Sirenian-specific residues shown in red shading. **e** Radioiodide uptake rates of human and sirenian SLC5A5 (NIS) constructs in HEK293T cells. The expression constructs human2dugong and dugong2human code for NIS proteins where the five sirenian-specific amino acids in (**c**) were replaced by their corresponding counterparts. The bar graph *y*-axis shows the percent of Na$^{125}$I uptake. Data derived from five independent experiments with single measurements (three for human2dugong), presented as mean ± S.D. relative to the control (empty *pcDNA3.1* vector), while dots represent individual data points per experiment (***$P < 0.0001$, ***$P < 0.001$, **$P < 0.01$, *$P < 0.05$, one-sided ANOVA with Tukey's multiple comparisons test). The exact *P* values compared to the control from left to right are $1.47 \times 10^{-3}$, 0, 0, $1.50 \times 10^{-6}$, and $1.90 \times 10^{-4}$; human compared to human2dugong, 0.015; and dugong compared to dugong2human, $1.00 \times 10^{-7}$. A *P* value of 0 indicates a number below $2.22 \times 10^{-16}$.

---

sirenians[23]. Consistent with previous reports[23–26], we observed a thin epidermis and a thick dermis in the dugong (Fig. 3a) and West Indian manatee (Fig. S4). We identified and validated, using dugong epidermis RNA-seq reads, the loss of multiple skin-associated genes (Fig. 3b and Supplementary Data 8). Notably, many of these genes are convergently lost in cetaceans, as revealed by manual literature searches and STRING[27] gene enrichment of the 15 shared sirenian pseudogenes identified in our analysis (Table S5 and Supplementary Note 5). In addition, our gene family screen revealed sirenian loss of late cornified envelope (LCE) gene family proteins expressed by the top layer of the epidermis (Fig. 3c). LCE gene numbers are also reduced in the afrotherian aardvark (*Orycteropus afer*; common ancestor ~80 Mya), suggesting a role in the evolution of its sparsely-haired skin[28].

## Daily activity patterns

Most terrestrial animals rely on circadian rhythmicity, a molecular clock regulated by the sun's daily cycle, to regulate activity patterns[29,30]. In contrast, many marine animals inhabiting coastal reef habitats and shallow waters rely heavily on the lunar (moon) cycle and its effect on water depth, food availability, and temperature[29]. The pineal gland is absent or non-functional in sirenians, cetaceans, and some terrestrial mammals. These species have lost genes associated with the synthesis and reception of melatonin; a hormone mediating light stimulation of the circadian clock[31–35].

Sirenians do not show a "classical" diel (24-h) activity pattern but exhibit short episodes of sleep during respiratory pauses, have unihemispheric slow wave sleep (i.e., one brain half is awake), and appear to respond behaviorally to tidal currents (tides may restrict foraging) and seasonal changes in water temperature[5,36–38]. We identified sirenian-specific amino acid substitutions in most core circadian clock genes (Fig. 1, Fig. 4a, and Supplementary Data 5), including numerous substitutions in all Period circadian regulator genes (five in *PER1*, 20 in *PER2*, and 13 in *PER3*). *PER2* is expressed by circadian pacemaker cells of the hypothalamic suprachiasmatic nucleus (SCN) and binds to cryptochromes (CRY) to regulate light-associated circadian rhythmicity and timing (including sleep patterns)[39]. A coimmunoprecipitation assay showed that dugong PER2 bound CRY1 better than wildtype human PER2 or human PER2 with a sirenian substitution (C1220P) in the CRY-binding domain (Fig. 4b, c and Fig. S5). Thus, at least one of the 19 other sirenian-specific PER2 residues may be required for the enhanced CRY-binding. Our analysis also revealed loss of *KCNK18* (also known as TRESK, TWIK-related spinal cord K⁺ channel) (Fig. 4d and Supplementary Data 8), a circadian clock-regulated ion channel. *KCNK18* is highly expressed in the SCN, and *Kcnk18⁻/⁻* mice cannot use light to differentiate between day and night[40].

## Gene loss and maladaptation in an altered environment

The loss of dozens of genes re-organized the skin of cetaceans[41] and sirenians (see above) over millions of years, allowing their semi-aquatic

ancestors to become fully aquatic. Climate change and human activities can outpace natural selection. Gene loss that may have been adaptive or tolerated in an ancestral environment can become maladaptive in a modern environment, especially in species with a long generation time[42]. Sirenians are long-lived (~70 years), with a generation time of ~20 years for the West Indian manatee and ~27 years for the dugong[5,43]. Threats to sirenians (mainly through loss of aquatic plant habitats) include pollution of waterways, fishing operations, and coastal dredging and reclamation—all of which may be exacerbated by changing climate patterns[44–46]. We identified three gene activation events that may be disadvantageous today: convergent loss of *PON1* and *CES3* in marine mammals (Supplementary Note 6 and Figs. S6 and S7) and sirenian-specific loss of *KCNK18*.

Shared and unique *KCNK18* inactivating mutations were observed in the dugong, Steller's sea cow, and West Indian manatee (Supplementary Data 8 and Fig. 4d). *KCNK18* loss was probably not inherently damaging (see Daily activity patterns section above) but may expose sirenians to natural and anthropogenic threats. Besides the hypothalamic SCN, skin sensory neurons express the gene, and *Kcnk18* knockout mice show elevated pain and avoidance behaviors when exposed to pyrethroid insecticides[47] or temperatures below 20 °C[48]. Given its geographic range (Fig. S1c), the Florida manatee (a subspecies of the West Indian manatee) manifests cold stress syndrome (CSS), a condition resulting from prolonged exposure to water temperatures below 20 °C characterized by multiple physiological changes and comorbidities of unknown genetic cause[5,49,50]. Although speculative, we hypothesize that loss of *KCNK18* decreases sirenian temperature tolerance (Fig. 4e) and that CSS is similar to semelparity in marsupials[51,52] in that a progressive and systemic deterioration of body condition and physiological function is mediated by an endocrine factor, perhaps from elevated levels of the stress hormone cortisol. Extant sirenians have relatively thin blubber compared to cetaceans and have lost *UCP1* (Supplementary Note 5), which could render them further susceptible to cold temperatures. Dugongs should arguably also be inherently sensitive to cold temperatures but have a more insulating integument, higher metabolic rate, and live in warmer waters throughout the year than Florida manatees[25,53]. The extinct Steller's sea cow related to the dugong further evolved a huge body size and thicker blubber to survive in the cold, sub-Arctic environment[54].

## A dugong whole-genome resequencing data set

We next considered the population genomics of dugongs from ten locations (Fig. 5a). To this end, we generated short-read whole-genome resequencing data from seven locations (skin biopsies of 99 individuals) spanning 2000 km of the Australian east coast from Torres Strait to Moreton Bay, Queensland (Supplementary Data 9). We obtained 3.46 Tb of data, with an average sequencing depth of 11.41×, and identified 71.25 million high-quality SNPs (average SNP density 24.61 SNPs/kb). Publicly available resequencing data (one individual

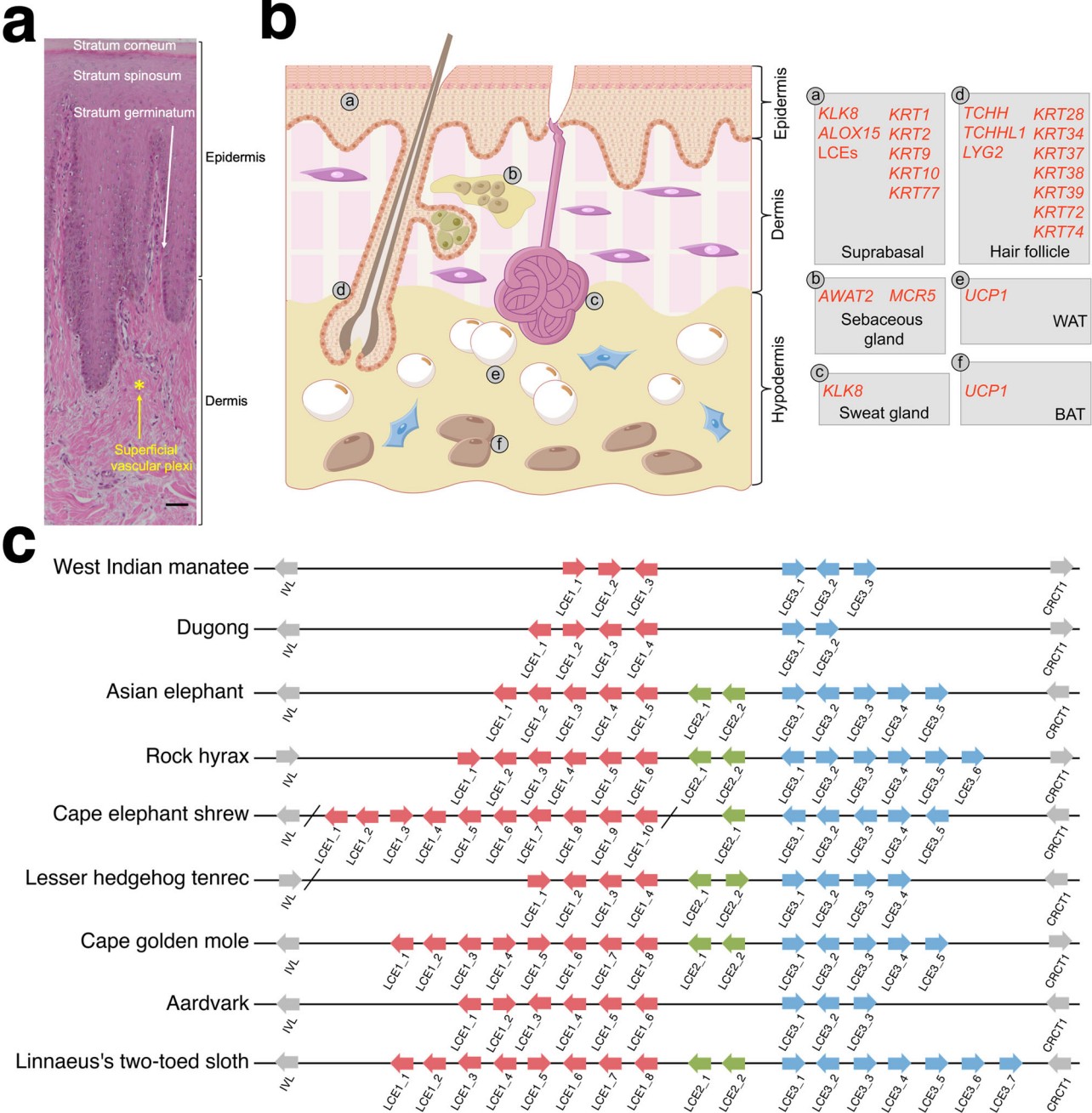

**Fig. 3 | Anatomy and gene changes of the sirenian integumentary system.**
**a** Representative histology of hematoxylin and eosin (H&E)-stained dorsal dugong skin. The scale bar represents 50 µm. **b** Schematic drawing of H&E-stained human skin (as a model of mammals) and associated genes with changes in sirenians. Left, overview of major skin anatomical structures: epidermis, dermis, hypodermis, and ectodermal appendages (hair follicles, sebaceous glands, and sweat glands). Right, genes and their site of expression. Inactivated genes are in red (see main text). WAT denotes white adipose tissue cell; BAT brown adipose tissue cell. **c** Schematic representation of late cornified envelope (LCE) gene clusters in afrotherian genomes. Arrows indicate genes and their direction of transcription; slashes indicate separate scaffolds; and different clusters are indicated by colored boxes.

each) was also obtained from two other Australian locations, Coogee Beach (New South Wales)[54] and Exmouth Gulf (Western Australia), and from waters off Okinawa (Japan). The average dugong genome-wide nucleotide diversity ($\pi$) and heterozygosity were $8.79 \times 10^{-4}$ and $8.80 \times 10^{-4}$, respectively, higher than those of the killer whale ($4.27 \times 10^{-4}$ and $3.54 \times 10^{-4}$)[55], northern elephant seal ($2.04 \times 10^{-4}$ and $1.78 \times 10^{-4}$)[56], and Indo-Pacific humpback dolphin ($1.55 \times 10^{-4}$ and $1.79 \times 10^{-4}$)[57]. The average heterozygosity of Moreton Bay individuals ($n = 32$) mirrored an estimate from a single individual from this locality[58] ($1.40 \times 10^{-3}$ vs. $1.60 \times 10^{-3}$).

### Population structure and differentiation

Principal component analysis (PCA) indicated that the individuals from Exmouth Gulf on the Australian west coast and Okinawa are genetically distinct to dugongs from the Australian east coast (Queensland and Coogee Beach) (Fig. 5b). The Coogee Beach individual clustered with Moreton Bay and Hervey Bay individuals. Because this individual stranded ~750 km from the accepted eastern Australian range during the summer (November), we propose it represents one of the few instances[59] of seasonal long-distance ranging from a population in close geographic proximity to Moreton Bay.

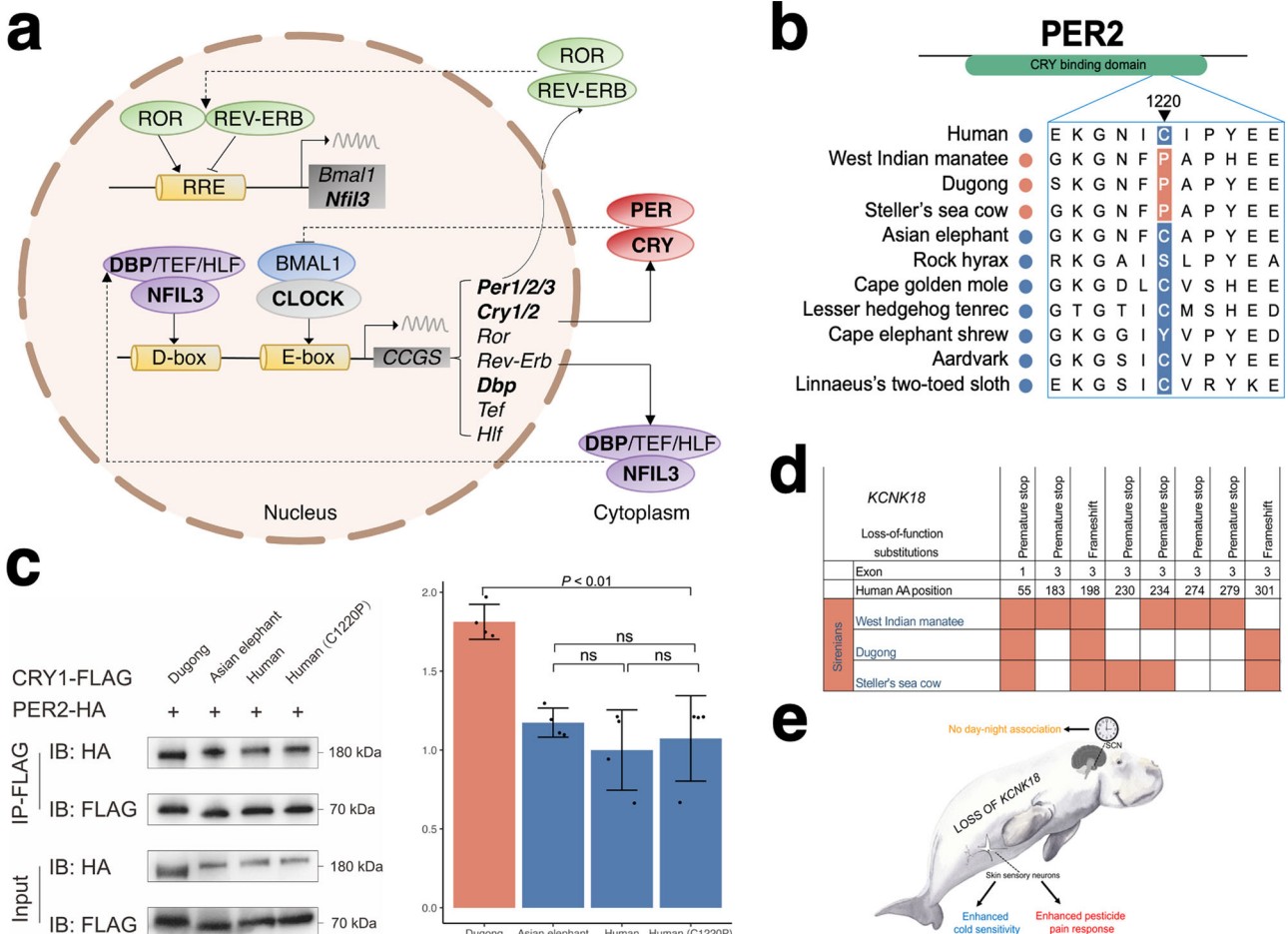

**Fig. 4 | Multiple gene changes may underlie the circadian activity of sirenians. a** Diagram of core circadian clock network. Genes with evolutionary changes in sirenians are bolded. D-box and E-box denote promoter elements found upstream of clock-controlled genes regulated by the transcription factors NFIL3 and CLOCK. **b** Alignment showing an amino acid substitution in the CRY binding domain of sirenian PER2. A sirenian-specific proline at residue 1220 is shown in red shading. **c** Left, a representative co-immunoprecipitation assay reveals a stronger binding ability of dugong PER2 with its CRY1 compared to Asian elephant and human forms. Human C1220P denotes human PER2 mutated to match dugong residue 1220. Numbers on the right side of the gels represent molecular weight marker locations (values in kDa). Full images of western blots in Fig. S5. Right, bar graph showing ImageJ densitometry of western blot from four independent experiments with single measurements, presented as mean arbitrary density units ± S.D. relative to human, while dots represent individual data points per experiment. One-sided ANOVA with Tukey's multiple comparisons test (ns denotes not significant, a $P$ value > 0.05). The exact $P$ values were dugong-Asian elephant, $3.31 \times 10^{-3}$; dugong-human, $4.48 \times 10^{-4}$; dugong-human (C1220P), $1.02 \times 10^{-3}$; Asian elephant-human, 0.62; Asian elephant-human (C1220P), 0.89; human–human (C1220P), 0.95. **d** Overview of inactivating substitutions in sirenian *KCNK18*. Colored cells indicate substitutions. **e** Illustration of anatomical sites where *KCNK18* plays important roles and potential functional outcomes of its loss in sirenians. SCN denotes the hypothalamic suprachiasmatic nucleus. Image of dugong courtesy of Liudmila Kopecka/Shutterstock.com.

Interrogation of our 99-individual Queensland data set showed pronounced genetic structure into a northern and a southern group since ~10.7 kya (95% CI: 9.1–12.2 kya), agreeing with a recently reported but undated potential ecological barrier at or near the Whitsunday Islands on the Great Barrier Reef[60] (Fig. 5a, c and Fig. S8). Summary statistics echoed the structure, revealing distinct genetic diversity (Fig. S9) and heterogeneity. The average pairwise fixation index ($F_{ST}$) values between the three northern and the four southern locations were around 0.1, indicating moderate differentiation (heterogeneity) between them. In contrast, there was no apparent within-group differentiation in the southern group (average $F_{ST}$ 0.023) (Fig. 5d). A TreeMix consensus tree (Fig. 5e) was largely concordant with population clustering inferred by PCA (Fig. 5b), STRUCTURE (Fig. 5c), and a Neighbor-Joining tree (Fig. S8). While the TreeMix topology (Fig. 5e) and Patterson's $D$-statistics (ABBA-BABA-test) (Fig. S10 and Supplementary Data 10) showed admixture between the northern and southern populations, the low weights of the migration events inferred by TreeMix (see ref. 61) may reflect gene flow before the emergence of

the north-south barrier ~10.7 kya. TreeMix and $D$-statistics showed no gene flow between Airlie Beach and the six other locations, indicating that dugongs at this location are genetically isolated.

Four complementary methods (XP-EHH[62], XP-CLR[63], π, and $F_{ST}$) were used to detect regions with putative selective sweeps, SNPs under selection. We identified a two-megabase region (24.48-26.50 Mb) on chromosome 18 under positive selection in the northern dugong group (Fig. 5f). This region contains five annotated protein-coding genes (Table S6). They comprise three immunoglobulin genes, the nuclear pore transporter *NUP42* (also known as *NUPL2*), and ClpX protease (*CLPX*). Among these, only *CLPX* SNPs cause amino acid residue changes (Ile197Thr) unique to the northern dugong group and have an ortholog in afrotherians and other mammals. The threonine at residue 197 (predicted as "benign" by PolyPhen-2 and "tolerated" by SIFT; Table S6) was not found in other sirenians and the 120 mammalian species in OrthoMaM (Fig. S11). CLPX is a mitochondrial protein required to synthesize oxygen-binding hemoglobin that buffers oxidative stress[64,65]. While it is difficult to conclude the driving force

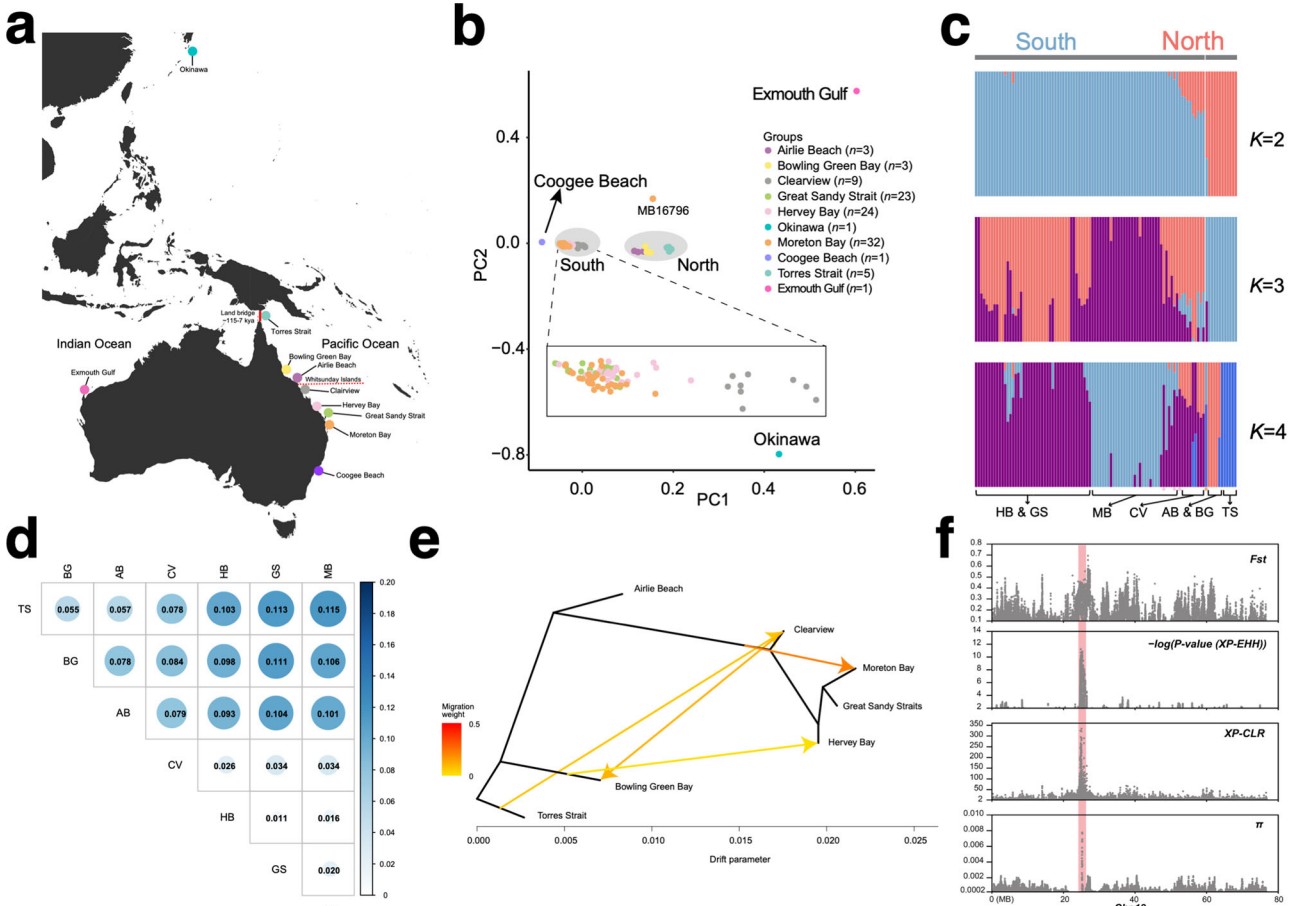

**Fig. 5 | Evidence of population structure by dugongs on the Australian east coast. a** Map indicating approximate sites of origin of re-sequenced dugongs. The red continuous line indicates a historical Torres Strait land border (115–7 kya); the dotted red line, a potential ecological barrier at or near the Whitsunday Islands on the Great Barrier Reef. Geographical locations are approximate. **b** Principal component (PC) analysis of the dugong individuals in (**a**). The gray highlighting shows clustering consistent with geographic locations south and north of the Whitsunday Islands also reported by ref. 60. **c** structure (ADMIXTURE) plots demonstrate substantial structure, supporting a break between population north and south of the Whitsunday Islands. Genetic structure inferred by varying the ancestry components (*K*) shown. **d** Pairwise population differentiation index ($F_{ST}$) differences between northern (TS, BG, and AB) and southern (CV, HB, GS, and MB)

population groups. Calculated in non-overlapping 10-kb sliding windows with 2-kb step sizes. **e** Gene flow among populations detected by TreeMix. A Maximum likelihood tree was inferred with four migration events. Terminal nodes are labeled by locality (see **a**). Arrows represent gene flow, and heat map colors reflect the intensity of gene flow, from low (yellow) to high (red). **f** A ~ 2 Mb selective sweep region on chromosome 18 identified by comparing population groups north and south of the Whitsunday Islands. Population differentiation ($F_{ST}$), normalized XP-EHH, XP-CLR, and relative nucleotide diversity (π) are plotted. The $-log(P value(XP-EHH))$ is the $log_{10}$ of a two-sided *P* value for testing the null hypothesis that no selection occurred, with a value ≥ 5 equivalent to a *P* value of 0.00001. Note the positive values, indicating selection acting on the northern dugong population.

behind the selective sweep of *CLPX* and any functional effect of its amino acid substitution in northern Queensland dugongs, sirenians are vulnerable to climate change and regional environmental conditions that dramatically alter their nearshore habitats (particularly seagrass growth)[45]. Northern Queensland has seen distinct climate change events after the formation of the ostensible ecological barrier at the Whitsunday Islands ~10.7 kya. The sea level on the north-east Queensland coast has continuously decreased since about 6000 years ago[66,67]. Climate change in the region has also seen increasing frequency and intensity of weather events affecting seagrass habitats— including high seasonal summer rainfall and coastal runoffs, and cyclones. Coastal bathymetry is also more variable in northern Queensland, with seagrasses found from inshore shallow estuaries through to reef flats and deeper subtidal inter-reefal areas[68].

## Demography of *Vulnerable* and recently extinct dugongs

Despite their numbers, the ~165,000 dugongs in Australian waters are listed as *Vulnerable* (at high risk of extinction in the medium-term future) by the IUCN[6,7]. The number of dugongs in other locations is

orders of magnitude lower. The dugong recently became functionally extinct in Chinese[69,70] and Japanese waters[71] and is at risk elsewhere in Asia, Oceania, and eastern Africa[7,72]. Pairwise sequential Markovian coalescent (PSMC)[73] analysis of dugong autosomes was used to track changes in effective population size ($N_e$; the number of individuals that will contribute to the next population) during the Pleistocene (about 2.6 Mya to 20 kya) (Fig. 6a and Fig. S12). All examined dugongs showed a $N_e$ decline in the mid-Pleistocene until ~400–500 kya that was also observed in a population of cold-resistant Steller's sea cow off the Arctic Commander Islands[54,74], suggesting that lower seawater temperatures and sirenian cold stress syndrome did not drive this dugong population decline. Individuals from the seven eastern Queensland locations had near-identical demographic histories. The Coogee Beach individual's PSMC curve mirrored the Queensland individuals, agreeing with the above PCA. The Exmouth Gulf individual from the Indian Ocean showed a distinct $N_e$ to the other Australian populations from the Pacific Ocean range but, nevertheless, a curve that likely reflects the broadly similar environmental conditions across the Australian continent (Fig. 6a). Our PSMC curve of Moreton Bay individuals (Fig. 6a

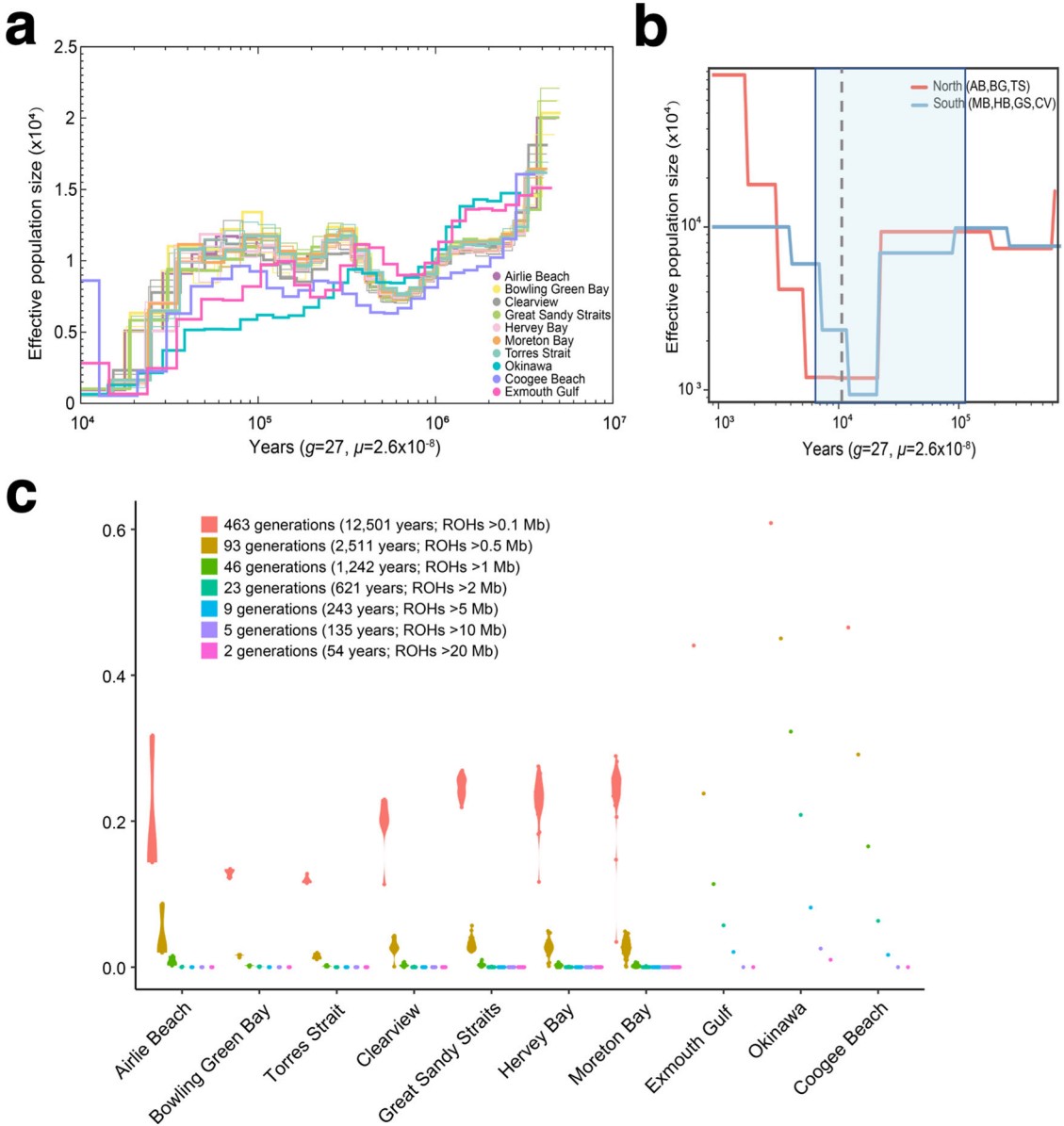

**Fig. 6 | Demographic history of dugongs. a** Demographic history of dugong individuals from 11 locations inferred using PSMC. Okinawa, Exmouth Gulf, and Coogee Beach are single individuals, while three individuals from the remaining Queensland locations are plotted. **b** Effective population size of populations from the Australian east coast (Queensland) inferred using SMC++. The most recent Torres Strait land border period is shaded in blue; the gray continuous line indicates the estimated emergence time of a potential ecological barrier at or near the Whitsunday Islands. **c** Inbreeding in dugong individuals from 463 to two generations ago. The *y*-axis shows $F_{ROH}$, the sum length of all ROHs (runs of homozygosity) of a particular length cutoff divided by the autosomal genome length; the *x*-axis indicates various dugong populations.

and Fig. S12) was similar to a recent study[58] that examined a single individual from this location, but the effective population size was smaller in our dataset (e.g., ~600,000 vs. ~12,000 individuals about 100,000 years ago). We speculate that the much higher $N_e$ in the recent study stems from different mutation rate parameters ($6.25 \times 10^{-9}$ vs. $2.60 \times 10^{-8}$ per site per generation in our study) or failure to remove chromosome X before PSMC analysis—a step that can influence effective population size estimates (see refs. [73,75]). In contrast to dugongs from Australian waters, the $N_e$ of the individual from Japanese waters declined continuously over the last 400,000 years. The effective population size differences of dugongs on the Queensland coast in the recent past (i.e., <20 kya) reconstructed using SMC++ (Fig. 6b) were consistent with the distinct genetic diversity and heterogeneity associated with their genetic break ~10.7 kya. We speculate that the northern dugong populations are more "genetically healthy"

because they resumed panmixia with populations from the Indian Ocean once sea levels rose and covered a Torres Strait land bridge present from ~115–7 kya (see Fig. 5a).

Runs of homozygosity (ROHs) in a genome reflect the level of inbreeding and can provide conservation management guidance[76,77]. Longer ROHs suggest recent inbreeding. The total length and number of ROHs were generally small and similar across the 99 dugongs sampled on the Queensland coast (Supplementary Note 7 and Fig. S9d–f), with no evidence of inbreeding in recent times (since 1242 years ago) (Fig. 6c and Supplementary Data 11). The individual from Exmouth Gulf ($F_{ROH>1Mb}$ = 0.021, nine ROHs spanning 60.1 Mb) showed evidence of inbreeding as recent as nine generations ago (243 years), consistent with a small or isolated population at this time. An illustration of how environmental events may affect population size in this locality is tropical cyclone Vance (1999) that appears to have resulted

in a large-scale emigration from Exmouth Gulf that reduced the dugong population from 1000 to less than 200 individuals in five years[78]. It follows that other migrations on a population scale in the last two centuries may have improved the genetic diversity of the Exmouth Gulf ancestors. The historical demography (PSMC) of the Okinawan individual aligned with its genome diversity estimates. It had one magnitude of order lower genome-wide heterozygosity ($5.65 \times 10^{-4}$) compared with Australian dugongs (~$1 \times 10^{-3}$). One-third of its genome was in ROH segments above one megabase (389 ROHs spanning 934.4 Mb), with evidence of inbreeding as recently as 135 years ago ($F_{ROH>10Mb} = 0.025$, four ROHs spanning 73.6 Mb) to 54 years ago ($F_{ROH>20Mb} = 0.010$, one ROH spanning 29.2 Mb). This pattern is consistent with an ancient population bottleneck and subsequent extensive inbreeding until recent times, mirroring a *Critically Endangered* Sumatran rhinoceros (*Dicerorhinus sumatrensis*) population on the Malay Peninsula[79]. Dugongs in Japanese waters were hunted for centuries, and the Okinawan dugong showed a further dramatic reduction in population size from the late 1900s until its likely functional extinction in 2019[71].

## Discussion

Using comparative genomic approaches, we here report genetic changes that may underlie sirenian features. Because the ancestors of extant manatees and the dugong diverged at crown Sirenia ~30 Mya, these changes were likely critical for their transition to an aquatic habitat. The lack of a pineal gland and their genetic background support that the circadian clock (i.e., sleep-wake cycle) of sirenians has been recalibrated (e.g., see ref. [38]), likely to facilitate an activity pattern in a more light-limited, fully aquatic environment heavily reliant on lunar tidal currents and water temperature fluctuations. We appreciate, however, that the function of sirenian-specific changes should be further assessed in vivo—as illustrated by a recent follow-up study on amino acid substitutions of panda DUOX2 in CRISPR-Cas9-edited mice[80]. This thyroid hormone synthesis-associated protein also has unique residues in sirenians. Many of the same genes of the integumentary system, particularly those expressed in the outer layers of the skin[32,81–87], were lost by both sirenians and cetaceans. Our data support the idea that convergent gene loss occurs in species with similar ecological pressures[88] (here, the transition to a fully aquatic lifestyle by distantly related species over ~50−60 My of evolution). We also observed gene losses that may be maladaptive in a modern environment, including the previously reported paraoxonase 1 gene (*PON1*)[89]. Sirenian *KCNK18* loss is possibly related to their shift in activity patterns and may render them susceptible to water temperatures affected by climate change and human activities. Its role in sirenian cold stress syndrome (CSS) should be investigated.

Our population genomics analysis offers insights into dugong diversity and demography. We conclude that while dugongs on the Australian east and west coasts have a genetic diversity comparable to other marine mammals of conservation concern, most populations (as indicated by our resequencing of 99 individuals from the Queensland coast) are in numbers that likely limit inbreeding. Viewed through a genetic lens, these populations can be considered relatively robust and healthy. In contrast, the dugong from the functionally extinct Okinawan population showed a continuous decline in effective population size and an ROH pattern consistent with extensive inbreeding for millennia and, likely, climatic ages. Recent human activities in the last 100 years, including overfishing and coastal development that reduced seagrass abundance, likely fast-tracked extinction of dugong populations in Japanese and Chinese waters and is a considerable risk factor elsewhere[46,69–71]. Future studies should interrogate whole-genome resequencing data from extant and extinct dugong populations worldwide—including modern, historical, and ancient samples. Such efforts promise to shed further light on dugong evolution and inform their conservation. We also confirm[60] and date a north-south genetic break that emerged approximately 10.7 thousand years ago on the Australian east coast and identify a two-megabase genetic sweep region that may be associated with historical and recurrent environmental differences between the north and south coast and formation of an ecologically distinct population group (i.e., ecotype[90]). Our data set allows further study of genetic structure related to geographical regions and environmental variables.

In conclusion, our study reveals insights into sirenian biology and the transition of terrestrial mammals to an aquatic lifestyle and provides a basis for future genomic explorations.

## Methods

### Sample collection and research ethics

Dugong samples were collected under the following permits issued to J.M.L.: The University of Queensland Animal Ethics Permits SBS/360/14, SBS/181/18, Scientific Purposes Permits WISP07255110 and WISP14654414, Moreton Bay Marine Parks Permit #QS2000 to #QS2010CV L228, Great Sandy Marine Parks Permit QS2010-GS043, and Great Barrier Reef Marine Park Permits #G07 = 23274:1 and G14/36987.1. All applicable institutional and/or national guidelines for the care and use of animals were followed.

Liver tissue (sample D201106) for reference genome sequencing was obtained by Queensland Parks and Wildlife Service from a recently deceased near-term female dugong fetus recovered from a cow that was hunted illegally in the Burrum Heads region of Hervey Bay Queensland in November 2020. The fetus was transported frozen to The University of Queensland and dissected by J.M.L. This fetal liver sample and a skin sample from a dugong (D110419) sampled after an indigenous subsistence hunt (see below for sampling details) in Torres Strait in 2011 were used for RNA-sequencing.

Skin samples for whole-genome resequencing were collected from 99 dugongs from seven geographic locations on the east coast of Australia: Airlie Beach (AB, $n = 3$), Bowling Green Bay (BG, $n = 3$), Clairview (CV, $n = 9$), Great Sandy Strait (GS, $n = 23$), Hervey Bay (HB, $n = 24$), Moreton Bay (MB, $n = 32$), and Torres Strait (D, $n = 5$; prefixed TS elsewhere in the manuscript) (Supplementary Data 9 and Fig. 5a). Briefly, skin was collected from the dorsum of wild free-swimming dugongs at each location, except Torres Strait, using a handheld scraper device[91]. In the Torres Strait, skin was excised from fresh dugong carcasses post-hunt by local Traditional Owners at Mabiuag Island. Skin samples were stored in salt-saturated 20% dimethyl sulfoxide (DSMO) and frozen at −20 °C until further processing for resequencing (see ref. [60]). Skin samples for histology were sub-sampled from the Torres Strait specimens and stored in 10% neutral buffered formalin until sectioned and stained.

Sampling and species distribution maps were generated using the R package "OpenStreetMap"[92].

**Genome sequencing.** High-molecular-weight DNA was extracted from a fetal liver sample (D201106) using a MagAttract HMW DNA Kit (QIAGEN). DNA quantity, purity and integrity were assessed by Qubit fluorometry (Invitrogen), Nanodrop spectrophotometry (Thermo Fisher Scientific), and pulse-field gel electrophoresis. Single-tube long fragment read (stLFR) libraries[93] were sequenced on an MGISEQ-2000 sequencer. A total of ~358 Gb (~100×) stLFR clean reads were generated after removing low-quality reads, PCR duplicates, and adaptors using SOAPnuke (v1.5)[94]. Hi-C libraries (Lieberman-Aiden et al.[95]) were prepared from the same fetal liver sample. Hi-C data (200 Gb 150 bp paired-end reads) were generated on the BGISEQ-500 platform.

**RNA sequencing.** RNA from fetal liver (sample D201106) and skin (sample D110419), extracted using an RNeasy Mini Kit (QIAGEN), was sequenced on the BGISEQ-500 platform to generate 86.7 and 96.3 Gb of 150-bp paired-end read RNA-seq data, respectively.

**Genome assembly.** The pipeline stLFRdenovo [https://github.com/BGI-biotools/stLFRdenovo], which is based on Supernova v2.11[96] and customized for stLFR data, was used to generate a de novo genome assembly. GapCloser v1.12[97] and clean stLFR reads (with the barcode removed with the stLFR_barcode_split tool [https://github.com/BGI-Qingdao/stLFR_barcode_split]) were used to fill gaps. Redundans v0.14a[98] was used to remove heterozygous contigs. Clean paired-end Hi-C reads validated by HiC-Pro v3.2.0_devel[99] were used to construct chromosome clusters with the 3D de novo assembly (3D DNA) pipeline v170123[100]. The assembly was further improved by interactive correction using Juicebox (v1.11.08)[101].

Assembly quality was assessed using BUSCO (Benchmarking Universal Single-Copy Orthologs) BUSCO v5.4.3[8], employing the gene predictor AUGUSTUS (v3.2.1)[102] and a 9,226-gene BUSCO mammalian lineage data set (mammalia_odb10).

**Genome annotation.** The sex of the sequenced individual (i.e., to determine if we could assemble the Y chromosome) was determined by visual inspection of the specimen prior to dissection, as well as BLAST[103] interrogation of African elephant[104] and published dugong[105] sex chromosome genes against our initial stLFR assembly and by comparing the mapping rate of clean stLFR sequencing reads against chromosome X and autosomes of the chromosome-level genome assembly (see the whole-genome resequencing section below for method). All approaches suggested that the sequenced individual was female.

We identified repetitive elements by integrating homology and de novo prediction data. Homology-based transposable elements (TE) annotations were obtained by interrogating a genome assembly with known repeats in the Repbase database v16.02[106] using RepeatMasker v4.0.5 (DNA-level)[107] and RepeatProteinMask (protein-level; implemented in RepeatMasker). De novo TE predictions were obtained using RepeatModeler v1.1.0.4[108] and LTRharvest v1.5.8[109] to generate database for a RepeatMasker run. Tandem Repeat Finder (v4.07)[110] was used to find tandem repeats (TRs) in the genome. A non-redundant repeat annotation set was obtained by combining the above data.

Protein-coding genes were annotated using homology-based prediction, de novo prediction, and RNA-seq-assisted (generated from fetal liver and skin) prediction methods. For homology-based prediction, protein sequences from five mammals were downloaded from NCBI: African bush elephant (*Loxodonta africana*) assembly LoxAfr3.0 [https://www.ncbi.nlm.nih.gov/datasets/genome/GCF_000785645.1]; Cape elephant shrew (*Elephantulus edwardii*) assembly EleEdw1.0 [https://www.ncbi.nlm.nih.gov/datasets/genome/GCF_000299155.1]; aardvark (*Orycteropus afer*) assembly OryAfe1.0 [https://www.ncbi.nlm.nih.gov/datasets/genome/GCF_000298275.1]; West Indian manatee (*Trichechus manatus latirostris*) assembly TriManLat1.0 [https://www.ncbi.nlm.nih.gov/datasets/genome/GCF_000243295.1], and human (*Homo sapiens*) assembly GRCh38.p12 [https://www.ncbi.nlm.nih.gov/datasets/genome/GCF_000001405.38]. These protein sequences were aligned to the repeat-masked genome using BLAT v0.36[111]. GeneWise v2.4.1[112] was employed to generate gene structures based on the alignments of proteins to a genome assembly. De novo gene prediction was performed using AUGUSTUS v3.2.3[113], GENSCAN v1.0[114], and GlimmerHMM v3.0.1[115] with a human training set. For RNA-seq-assisted gene prediction, 150 bp PE reads from fetal liver and skin, generated on an BGI-SEQ 500 instrument, were filtered using Flexbar v3.4.0[116,117] with default settings (removes reads with any uncalled bases). Any residual ribosomal RNA reads [the majority ostensibly removed by poly(A) selection prior to sequencing library generation] were removed using SortMeRNA v2.1b[118] against the SILVA v119 ribosomal database[119]. Transcriptome data (clean reads) were mapped to the assembled genome using HISAT2 v2.1.0[120] and SAMtools v1.9[121], and coding regions were predicted using TransDecoder v5.5.0[122,123]. A final, non-redundant reference gene set was generated by merging the three annotated gene sets using EvidenceModeler v1.1.1 (EVM)[124]. The gene models were translated into amino acid sequences and used in local BLASTp[103] searches against the public databases Kyoto Encyclopedia of Genes and Genomes (KEGG; v89.1)[125], NCBI non-redundant protein sequences (NR; v20170924)[126], Swiss-Prot (release-2018_07)[127], TrEMBL (Translation of EMBL [nucleotide sequences that are not in Swiss-Prot]; release-2018_07)[128], and InterPro (v69.0)[129]. The gene set was also examined using BUSCO v5.4.3 and its mammalia_odb10 gene set ('transcriptome mode').

**Phylogeny and divergence time estimation.** Genome and protein sequences of eight afrotherians and Linnaeus's two-toed sloth *Choloepus didactylus* (outgroup) (see Table S2) were downloaded from the NCBI or DNA Zoo databases.

We identified single-copy gene family orthologs using Ortho-Finder (v2.5.4)[130,131]. The coding sequences (CDS) for each species were aligned using PRANK v70427[132,133] and filtered by Gblocks v0.91b[134] to identify conserved blocks (removing gaps, ambiguous sites, and excluding alignments less than 300 bp in size). Finally, 7695 single-copy genes were concatenated into supergenes for phylogenetic analyses.

To identify conserved non-exonic elements (CNEEs), we generated whole-genome alignments (WGAs) using LASTZ v1.04.22[135] with the parameters "H = 2000 Y = 3400 L = 3000 K = 2400" and our dugong reference genome (Ddugon_BGI) as the reference. We then merged aligned sequences using MULTIZ (v11.2)[136]. To estimate the non-conserved model, we employed phyloFit (v1.4) in the PHAST package[137] with 4d sites in the Afrotheria alignments and the topology using default parameters. We ran phastCons with the non-conserved model to estimate conserved regions with the parameters "target-coverage = 0.3, expected-length = 45, rho = 0.31". Exon regions were excluded from the highly conserved elements to generate 627,279 CNEEs with a total length of 103.9 M longer than or equal to 50 bp.

Mitochondrial genomes and protein-coding sequences for the following species were obtained from NCBI GenBank: dugong (NC_003314.1), West Indian manatee (MN105083.1), Asian elephant (*Elephas maximus*; NC_005129.2), rock hyrax (*Procavia capensis*; NC_004919.1), lesser hedgehog tenrec (*Echinops telfairi*; NC_002631.2), golden mole (*Chrysochloris asiatica*; NC_004920.1), aardvark (NC_002078.1), Cape elephant shrew (NC_041486.1), and Linnaeus's two-toed sloth (NC_006924.1). We used MARS (Multiple circular sequence Alignment using Refined Sequences)[138] to rotate the mitochondrial sequences to the same origin as the dugong.

Separate maximum-likelihood (ML) phylogenetic trees of eight afrotherians and Linnaeus's two-toed sloth (outgroup) were generated with RaxML v8.2.9[139] (1000 bootstrap iterations) using coding sequences from 7695 genes, 1,127,156 fourfold degenerate sites in the 7695 genes, 5508 single-copy Benchmarking Universal Single-Copy Ortholog (BUSCO) genes[8,140], 627,279 conserved non-exonic elements (CNEEs), mitochondrial genomes, and 13 mitochondrial coding sequences. The resulting tree with the highest GTRGAMMA likelihood score was selected as the best tree.

We also used ASTRAL-III v5.6.2[141] to generate a coalescent species tree from non-overlapping 20 kb windows (to minimize linkage between subsequent windows from the WGAs generated above (see ref. 142)). Briefly, after excluding windows from alignments less than 2000-bp in size and with more than 25% gaps, 152,478 windows were used to generate window-based gene trees (WGTs) using RaxML (with 1000 bootstraps). WGTs with mean bootstrap support ≥80% were input into ASTRAL-III (with default parameters) to estimate an unrooted species tree. Apart from the mitochondrial trees (its whole genome and protein coding gene trees were not identical), the nuclear genome-derived ML trees showed the same topology as the ASTRAL WGT tree.

Divergence times between species was estimated using MCMCTree (a Bayesian molecular clock model implemented in PAML

v4.8[143]) with the JC69 nucleotide substitution model, and the concatenated whole-CDS supergenes as inputs. We used 100,000 iterations after a burn-in of 10,000 iterations. MCMCTree calibration points (million years ago; Mya) were obtained from TimeTree[144]: Cape golden mole-lesser hedgehog tenrec (58.4–81.7 Mya), Cape golden mole-Cape elephant shrew (63.0–87.5 Mya), Cape golden mole-aardvark (54.9–89.3 Mya), Cape golden mole-West Indian manatee (76.0–80.8 Mya), Linnaeus's two-toed sloth-West Indian manatee (84.0–97.9 Mya). We also included data from a recent manuscript[2] that employed a total-evidence approach (i.e., incorporating morphological, molecular, temporal, and geographic data from living and fossil species) to estimate a divergence of Trichechidae (i.e., West Indian manatee ancestor) from Dugongidae (i.e., dugong) ancestors 31.7–36.7 Mya.

To further examine the relationship between paenungulates, we also considered retroelements. Retroelements (i.e., LINEs, SINEs, and LTRs) are considered near homoplasy-free markers given their insertion mode[145–148]. We employed a recently developed pipeline[147] that requires the 2000 bp flanking a retroelement insertion site to assign informative phylogenetic markers from pairwise whole-genome alignments (here, with Ddugong_BGI as the reference genome). The Kuritzin-Kischka-Schmitz-Churakov (KKSC) test[149] was used to assess presence/absence matrixes.

**Comparative genomics analysis strategy.** To summarize our analysis strategy (see detailed methods below) and manuscript data, we first compared signatures of natural selection with literature searches (comprehensive reviews on the anatomical and physiological adaptations of sirenians to aquatic life, including refs. 5,150) to discover broad functional categories associated with sirenian adaptations. Enrichment analysis of positively selected (Supplementary Data 1 and 2) and rapidly evolving (Supplementary Data 3 and 4) genes using KOBAS revealed an over-representation (Benjamini–Hochberg $P < 0.05$) of terms related to thyroid hormone synthesis, the cardiovascular system, integumentary system (i.e., cornified envelope), and circadian activity. Sirenian-specific amino acid substitutions in the thyroid hormone pathway and circadian clock proteins were next identified using FasParser2[151,152] (Supplementary Data 5) and validated against our genome resequencing data set of 99 dugongs, the 120 mammalian species in OrthoMaM[153], and by BLAST[103] searches of NCBI and Ensembl databases. Functional in vitro assays were used to evaluate selected substitutions. CAFE[154] revealed loss of gene families of the integumentary (i.e., cornification/keratinization) and olfactory systems (Supplementary Data 6 and 7). A recently described pipeline[155] confirmed reported pseudogenes (e.g., refs. 89,32) among the 15 shared by sirenians (Supplementary Data 8 and Table S5) but also gene inactivation events not previously described.

**Gene family analysis.** Gene family expansion and contraction analysis was performed using CAFE v4.2[154] with our consensus phylogenetic tree (also see Fig. S1c) as the input. Expanded and contracted gene families on each branch of the tree were detected by comparing the cluster size of each branch with the maximum-likelihood cluster size of the ancestral node leading to that branch. A smaller ancestral node indicates gene family expansion, whilst a larger ancestral node indicates family contraction. Gene families with a $P$ value $< 0.01$ were defined as significantly expanded or contracted in a branch of interest.

**Sirenian gene selection.** Selection signatures of the 7695 single-copy gene family orthologs in our nine-species data set were identified using their coding sequences and PAML codeml v4.8[143].

We tested for positively selected genes (PSGs) on sirenian branches by comparing branch-site models, allowing a codon site class with (dN/dS; also known as omega, ω) > 1 along foreground branches, with branch-site null models. We identified sites under positive selection

using Bayes Empirical Bayes (BEB) in PAML[156] and a Benjamini–Hochberg $P$ value cut-off set at 0.05.

Rapidly evolving genes [REGs, i.e., genes with an elevated dN/dS] in sirenians were identified using the PAML branch model. The two-ratio model (model = 2) allows one ratio for background branches and another for foreground (sirenians) branches, while the one-ratio model (model = 0) enforces the same ratio for all branches. Genes with a $P$ value (computed using the $\chi^2$ statistic) less than 0.05 and a higher ω value in the foreground lineage were considered REGs.

We employed KOBAS v3.0 [http://kobas.cbi.pku.edu.cn][157] gene enrichment with a Benjamini–Hochberg $P$ value cut-off set at 0.05 to identify functional categories that may underlie aquatic specializations of sirenians. Gene sets were also interrogated using STRING v12.0 [https://string-db.org][27], which includes "Reference publications" (i.e., publications with PubMed abstracts up to August 2022 and the PMC Open Access Subset up to April 2022) (false discovery rate cut-off set at 0.05).

**Gene loss in the Sirenian lineage.** To screen for gene loss in sirenians, defined as genes harboring premature stop codons and/or frameshifts in a species, we employed a previously published approach[155]. Briefly, the longest human protein sequence for each gene was mapped to genomes of the dugong, West Indian manatee, rock hyrax, and Asian elephant using BLAT v36[111] and genBlastA v1.0.1[158]. Next, the mapped genomic regions and 1000 bp upstream and downstream were examined for disruptions (nonsense mutations and frameshifts) to the gene coding sequences using GeneWise v2.4.1[112]. We removed loci hits belonging to large gene families (including olfactory receptors, zinc finger proteins, and vomeronasal receptors) and predicted proteins or intronless cDNA/expressed sequences[159]. False positives with disruptive mutations introduced by GeneWise, sequencing errors, or annotation errors were removed following the steps in ref. 155. Candidate pseudogenes with multiple disruptions were manually inspected to remove short or low-quality alignments. We also interrogated raw sequencing reads from dugong and West Indian manatee using BLAST[103] to confirm each disruption. Gene enrichment analysis was performed using KOBAS and STRING, as outlined above.

**Sirenian lineage-specific amino acid changes.** Amino acid alignments of the 7695 single-copy orthologs in our nine-species data set (eight afrotherians and Linnaeus's two-toed sloth) and FasParser v2[151,152] was used to identify amino acid residues specific to the sirenian lineage. Putative sirenian-specific residues of interest were further validated using 120 mammalian sequences downloaded from OrthoMaM v10c[153] (available at FigShare [https://doi.org/10.6084/m9.figshare.23975559]), as well as by interrogating NCBI and Ensembl databases using BLAST. Potential functional effects of substitutions were predicted by PolyPhen-2[160,161] and SIFT v6.2.1[162,163]. PolyPhen-2 predicts the possible impact of amino acid substitutions on the stability and function of human proteins using structural and comparative evolutionary considerations. SIFT predicts the potential impact of amino acid substitutions or indels on protein function.

**NIS radioiodide uptake assay.** DNA sequences containing the protein-coding region of dugong, manatee, and human *SLC5A5* (NIS), as well as dugong (L142M, G203W, A321P, A445V, and S539T) and human (M142L, W203G, P321A, V445A, and T539S) sequences where five amino acid substitutions were changed to their reciprocal residues, were synthesized by GenScript Biotech. Each DNA was individually subcloned into the *pcDNA3.1* vector (Invitrogen) to generate NIS plasmids. All expression constructs were sequenced to verify their nucleotide sequences. HEK293T cells were cultured in 6-well plates until reaching 50–60% confluency. Cells were transfected with 2 μg NIS plasmid using the PEI 40 K Transfection Reagent (Servicebio). Empty *pcDNA3.1* vector was used as a control. One day after transfection,

HEK293T cells were seeded into 24-well plates and treated with radiopharmaceuticals after 24 h. Briefly, cells were incubated with Na$^{125}$I (Shanghai Xinke Pharmaceutical Co.) for 1 h. The Na$^{125}$I added to each well of the culture plate was counted using a radioactivity meter (FJ-391A4) to obtain the total radioactive count T (μCi) per well. After discarding the radioactive supernatant, cells were washed twice with PBS solution. Radioactive Na$^{125}$I absorbed by the HEK293T cells was denoted the cell radioactive count C (μCi). The radioiodide uptake rate is shown as C/T%.

**Coimmunoprecipitation and immunoblotting of PER2 and CRY1.** DNA sequences containing the protein-coding region of dugong, Asian elephant, and human *PER2* and *CRY1* were synthesized by GenScript Biotech. In addition, a human *PER2* sequence with a sirenian-specific proline at residue 1220 (C1220) was synthesized. The CRY1 sequences contained a C-terminal 3×FLAG tag, the *PER2* sequences a C-terminal human influenza hemagglutinin (HA) tag. Each of the seven DNA sequences was individually subcloned into the *pcDNA3.1* vector (Invitrogen) and sequenced to confirm their identity. HEK293T cells were cultured in 10-cm Petri dishes until reaching 70–80% confluency and then transfected with 4 μg *CRY1-3×FLAG-pcDNA3.1* and 6 μg *PER2-3×HA-pcDNA3.1* using the PEI 40 K Transfection Reagent (Servicebio). Two days after transfection, cells were lysed in Western IP Cell Lysis buffer (Beyotime) supplemented with 1 mM PMSF (Biosharp) and subjected to coimmunoprecipitation. In total, 10% of the cell extracts were retained for input. Cell lysates were incubated with Anti-FLAG M2 Magnetic Beads (Sigma) at 4 °C overnight. After washing three times, the precipitates were resuspended in SDS−PAGE sample buffer, boiled for 5 min, and run on a 6% SDS−PAGE gel. Immunoblotting was performed using mouse monoclonal anti-HA (Proteintech cat. no 66006-2-Ig at 1:50,000 dilution) or anti-FLAG (Proteintech cat. no 66008-4-Ig at 1:25000 dilution) antibodies, and an anti-mouse secondary antibody (Proteintech cat. no SA00001-1 at 1:6000 dilution). A StarRuler broad-range (10−180 kDa) molecular weight marker (GenStar cat. no M221) was co-run to estimate protein weights.

**Whole-genome resequencing.** DNA from 99 dugongs sampled on the Australian east coast (see "Sample collection and research ethics") was extracted using a QIAamp DNA Mini Kit (QIAGEN) and sequenced on a DNBSEQ-G400 RS instrument by BGI-Australia to generate 100-bp paired end reads. We also obtained public sequencing data from Okinawa (Japan; DRR251525; sampled 17 November 2019), Coogee Beach (New South Wales, Australia; ERR5621402; sampled 25 November 2009)[54], and Exmouth Gulf (Western Australia, Australia; SRR17870680; sampled 3 June 2018). We obtained 101-bp paired-end WGS reads generated on the Illumina HiSeq 2000 platform (NCBI SRA SRR331137, SRR331139, and SRR331142) from a female West Indian manatee (*Lorelei*, born in captivity in Florida, USA)[164]. The manatee reads were employed as the outgroup in population genomics analyses. Raw data were filtered using SOAPnuke v2.1.5[94] to remove adapters and low-quality reads. For comparative analyses, all samples were down-sampled to ~10× coverage using SAMtools v1.9[121].

Because we had a chromosome-level dugong genome, we assigned sex to samples by using the Rx ratio method, where sequencing reads are mapped to a genome with an assigned X chromosome and the number of reads mapping to autosomes are compared to the X-chromosome (normalized by chromosome length)[165,166]. The Rx ratio approximates 1.0 for females and 0.5 for males. Briefly, we first identified the X chromosome of the West Indian manatee (assembly TriManLat1.0_HiC; chromosome-level genome based on an assembly reported by ref. 164) and dugong (assembly Dugong_BGI) by BLAST searches using the coding sequences of genes evenly distributed across the X chromosome of the African savanna elephant (*Loxodonta africana*)[104] and dugong Y chromosome genes[105]. This effort revealed that the ~167 Mb HiC_scaffold_7 and the ~149 Mb chr7 in West Indian manatee and dugong correspond to their respective X chromosome. Next, we aligned reads to the chromosome-level genomes using bowtie2 v2.3.4.3[167,168] (parameter: −no-unal to only retained mapping reads), followed by conversion to a BAM file and filtering using SAMtools v1.7[121,169] (removal of PCR duplicate and retaining reads with a quality score, *Q*, above 30). Index statistics for BAM files were generated using idxstats in SAMtools and parsed by modifying an R script available via ref. 165. Average sequencing depth was estimated from indexed and sorted BAM files using mosdepth v0.3.3[170] (parameters: -n −fast-mode −by 500). See Supplementary Data 9 for sample statistics.

**Identification and characterization of SNPs.** Including the X chromosome can interfere with demographic history estimates[73,171,172] and other population genomics analyses[173,174] (X chromosome SNP effects). Therefore, reads that could be mapped to the X chromosome were removed using bowtie2 (v2-2.2.5)[168] with default parameters, resulting in ~34.14 Tb of clean data with an average sequencing depth of around 11.41-fold. The filtered clean reads were aligned to our Dugong_BGI reference genome using BWA v0.7.12-r1039[175] with default parameters. SAMtools v1.2[176] was employed to convert SAM files to BAM format and to sort alignments, followed by the Picard package v1.54 [https://broadinstitute.github.io/picard] for duplicate removal. Next, GATK v4.1.2.0[177] was utilized to realign reads around InDels and detect SNPs. Briefly, we obtained the genomic variant call format (GVCF) in ERC mode based on read mapping with the parameters "-T HaplotypeCaller, -stand-call-conf 30.0 -ERC GVCF". Joint variant calling was then conducted with the GATK *CombineGVCFs* module. Lastly, the GATK's *VariantFiltration* module was used for hard filtering with the parameters "−filter-name LowQualFilter −filterExpression QD < 2.0 || MQ < 40.0 || FS > 60.0 || ReadPosRankSum < −8.0 || MQRankSum < −12.5 || SOR > 3.0", as recommended by GATK. This process generated 61,741,769 SNPs.

**Analysis of population structure.** To quantify the genetic structure of dugong populations, we first carried out SNPs filtration using vcftools v0.1.16[178] with the parameters "-max-missing 0.95". Plink (v1.90b6.6)[179] was used to perform SNP quality control with the parameters "-geno 0.1 −maf 0.01". Linkage disequilibrium was also used as a criterion to filter SNPs for downstream analysis using Plink with the parameters "-indep-pairwise 50 5 0.2". Next, ADMIXTURE v1.3.0[180], with the parameters "−cv -j20 -B5" for multiple repeats, was used to perform ancestry inference. To construct the population evolutionary tree, an identity by state (IBS) distance matrix was constructed using Plink with the parameters "−distance 1-ibs" MEGA7[181] was used to construct the Neighbor-Joining Tree (NJ tree) based on the IBS matrix, and the evolutionary tree was visualized using the Interactive Tree Of Life (iTOL) online tool v6 [https://itol.embl.de][182]. In addition, PCA (principal component analysis) was performed using Plink with the parameters "−make-rel −pca 3" and visualized by the R ggplot2 package[183]. The divergence time between population groups was estimated using dadi v2.1[184].

**Estimation of genome heterozygosity and runs of homozygosity.** To estimate the heterozygosity of each dugong individual, we used the Plink function "−het" to detect heterozygous SNPs from the final SNP data set. Additionally, we estimated runs of homozygosity (ROH) using the R package *detectRUNS* [https://cran.r-project.org/web/packages/detectRUNS/vignettes/detectRUNS.vignette.html] with the parameters "windowSize = 50, minSNP = 30, maxGap = 1,000,000, minLengthBps = 100,000, minDensity = 1/100,000" based on the same filtered SNPs set used in the "population structure analysis" section.

ROH's per dugong generation was calculated as follows. FASTEPRR v2[185] was used to estimate the recombination rate of the dugong population based on the 99 Queensland individuals. It was estimated to be ~1.08 cM/Mb. The coalescent times of ROH for each

subpopulation were calculated as $g = 100/(2rL)$ (see ref. [186]), where $g$ is the expected time (in generations; 27 years[43]) back to the parental common ancestor, $r$ is the recombination rate and $L$ is the length of the ROH in megabases. Thus, ROHs <100 kb in length are estimated to result from inbreeding more than 12,501 years ago; ROHs >100 kb, up to 12,501 years ago (463 generations); ROHs >500 kb, up to 2511 years ago (93 generations); ROHs >1 Mb, up to 1242 years ago (46 generations); ROHs >2 Mb, up to 621 years ago (23 generations); ROHs >5 Mb, up to 243 years ago (9 generations); ROHs >10 Mb, up to 135 years ago (5 generations); and ROHs >20 Mb, up to 54 years ago (2 generations).

**Identification of selective sweep regions.** Four different methods, including $F_{ST}$ (population differentiation), $\pi$ (relative nucleotide diversity), XP-EHH (the cross-population extended haplotype homozygosity statistic)[62], and XP-CLR (the cross-population composite likelihood ratio test)[63] were employed to assess selective sweeps between the northern and southern Queensland groups. Briefly, vcftools v0.1.16[178] with the parameters "–max-missing 0.95 –maf 0.01" was used to perform autosome SNPs quality control. Next, the pairwise fixation index ($F_{ST}$) was calculated between the seven Queensland locations and between the whole northern (Torres Strait, Bowling Green Bay, and Airlie Beach) and southern (Moreton Bay, Great Sandy Straits, Hervey Bay, and Clairview) groups from Queensland using vcftools with the parameter "–fst-window-size 10000 –fst-window-step 2000". $\pi$ was calculated for each of the seven groups and the whole northern and southern groups using vcftools with the parameters "–window-pi 10000 –window-pi-step 2000". XP-CLR scores were calculated using xpclr v1.1.2[63] with default parameters. Because XP-EHH requires the genetic distance between adjacent SNPs, we considered a chromosome segment of 1 Mb to be 1 cM. The filtered SNPs were phased using "vcf_phase.py", with the parameter "–phase-algorithm beagle" from the PPP (v0.1.12) pipeline[187], followed by vcftools with the parameter "–IMPUTE". Next, the *xpehhbin* module from Hapbin v1.3.0[188] was used to calculate XP-EHH values (see ref. [189]). The XP-EHH scores were normalized and corresponding $P$ values were calculated. If a $P$ value was less than 0.01, we considered the region a candidate sweep region. An XP-EHH score is directional: a positive score suggests that selection occurred in the northern group; a negative score, the southern group. Genomic regions that overlapped between all four methods were considered candidate selective sweep regions.

**Demographic history inference.** Generation time ($g$) and mutation ($\mu$) rate are necessary to infer the demographic history of populations. The estimated generation time of the dugong (~27 years) has been reported previously[43]. We estimated the mutation rate for dugong using r8s[190] and the single copy orthologous genes described in the Phylogeny and divergence time estimation section. The final estimated mutation rate was $2.6 \times 10^{-8}$ per site per generation.

Pairwise sequentially Markovian coalescent (PSMC) v0.6.5-r67[73] was employed to infer historical effective population size ($Ne$) fluctuations in dugongs. PSMC can quantively reveal changes in $N_e$ from approximately 1 million to 20 thousand years ago[73]. We first constructed diploid consensus sequences for each sample using SAMtools v1.9[176] mpileup and BCFtools v1.4[169] with the parameters "-C50" and "-d 4 -D 24". The consensus sequences were transformed to PSMC input format using fq2psmcfa with the parameter "-q20". Finally, PSMC was used to infer the population history with the parameters "-N25 -t15 -r5 -p 4 + 25∗2 + 4 + 6" and 100 rounds of bootstrapping.

Because of the insufficient resolution of PSMC in estimating demography more recently than ~20 kya[191], we employed SMC++ v1.15.5[192] to infer more recent population history for Queensland dugong individuals (see "Analysis of population structure" above). The SMC++ modules *vcf2smc*, *estimate*, and *plot* were used. One of the 32 individuals sampled from Moreton Bay (MB16796; an older female) clustered with the northern group in a PCA (Fig. 5b) and a neighbor-

joining tree (Fig. S8). It may reflect low-level individualistic movement rather than population migration (see refs. [60,193,194]) between northern and southern populations and was excluded from the demographic history analysis.

To measure gene flow (i.e., migration) between populations, we interrogated our SNP data set with Dsuite (across all dugong populations with the West Indian manatee as the outgroup)[195] to obtain Patterson's $D$ (ABBA-BABA statistic) and TreeMix v1.13[61] to visualize gene flow (migration) on a maximum likelihood tree of populations. TreeView was run with the parameters "-root TS -k 500000 -m 0-10". Torres Strait, TS, was used to root (parameter *-root*) the tree, SNPs were grouped in 500,000-bp windows (parameter *k*). We estimated the optimal number of migration events (parameter *m*) for TreeMix analysis using the R package optM[196].

### Reporting summary
Further information on research design is available in the Nature Portfolio Reporting Summary linked to this article.

### Data availability
Dugong sequencing reads (including stLFR, Hi-C, RNA-seq, resequencing) and the Ddugon_BGI genome assembly are available at NCBI BioProject PRJNA1114306. Dugong SNP data in VCF format are available at the European Nucleotide Archive (ENA) and linked to the NCBI BioProject. Multiple sequence alignments (MSAs) of thyroid hormone pathway and circadian clock genes with sirenian-specific amino acid substitutions are available on FigShare [https://doi.org/10.6084/m9.figshare.23975559]. Public datasets used in this study are available from NCBI RefSeq, NCBI SRA, and DNA Zoo. Table S2 lists afrotherian nuclear genome assemblies used in protein ortholog prediction and whole-genome alignments. Linnaeus's two-toed sloth was used as an outgroup species (NCBI assembly mChoDid1.pri). Dugong genome annotation employed genes from the NCBI assemblies of West Indian manatee (ASM3001377v1), African bush elephant (LoxAfr3.0), Cape elephant shrew (EleEdw1.0), aardvark (OryAfe1.0), and human (GRCh38.p12). Mitochondrial genome assemblies and protein-coding sequences were obtained from NCBI dugong (NC_003314.1), West Indian manatee (MN105083.1), Asian elephant (NC_005129.2), rock hyrax (NC_004919.1), lesser hedgehog tenrec (NC_002631.2), golden mole (NC_004920.1), aardvark (NC_002078.1), Cape elephant shrew (NC_041486.1), and Linnaeus's two-toed sloth (NC_006924.1). Gene loss was validated using NCBI SRA reads from West Indian manatee (SRR8616893, SRR331138, SRR24090881, SRR24090877, and SRR24090880) and Steller's sea cow (ERR5559486, SRR12067494, and SRR12067500). Population genomic analyses employed SRA reads from dugong (DRR251525, ERR5621402, and SRR17870680) and West Indian manatee (SRR331137, SRR331139, and SRR331142).

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

## Acknowledgements

We thank those who assisted with the Australian dugong tissue collection, including The University of Queensland Dugong Team (especially Alex McGowan, Erin Neal, and Rob Slade) members of the Mabuiag Island community of Torres Strait (especially Terrence Whap), and Steve Hoseck (Southern Marine Parks, Queensland Parks and Wildlife) for facilitating access to the dugong fetus used for stLFR and RNA-sequencing. We also thank the laboratory of Prof. Kai Yang at Soochow University (China) for their radioisotope expertise and access to equipment, Dr. Erina Young at Murdoch University (Australia) for information on the dugong individual from Exmouth Gulf, Western Australia, Dr. Shaohong Feng (BGI Research) for access to scripts associated with their retroelement phylogenetic marker pipeline[147], and Prof. Harold H. Zakon (The University of Texas at Austin) for helpful feedback on the revised manuscript. Unpublished genome assemblies and sequencing data for the West Indian manatee, Asian elephant, rock hyrax, and aardvark were used with permission from the DNA Zoo Consortium [https://www.dnazoo.org]. Support for this research was provided by the Chinese Ministry of Science and Technology National Key Programme of Research and Development (grant 2022YFF1301601 to R.T. and S.L.), the National Natural Science Foundation of China (grant 42225604 to S.L. and grant 32270441 to R.T.), the Young Elite Scientists Sponsorship Program of the China Association for Science and Technology (grant 2023QNRC001 to R.T.), the Sea World Foundation (Australia) and Winifred Violet Scott Foundation (to J.M.L.), the "One Belt and One Road" Science and Technology Co-operation Special Program of the International Partnership Program of the Chinese Academy of Sciences (grant 183446KYSB20200016 to S.L.), the Specially-appointed Professor Program of Jiangsu Province (to I.S.), the Jiangsu Foreign Expert Bureau (to I.S.), and the Jiangsu Provincial Department of Technology (grant JSSCTD202142 to I.S.).

## Author contributions

R.T., J.M.L., G.F., S.L., and I.S. conceived the study. J.M.L. and H.L.S. collected or curated samples/specimens. R.T., Z.J., and L.W. performed laboratory work. R.T., Y.Z., H.K., F.Z., J.W., and I.S. performed computational biology work. R.T., S.L., G.F., and I.S. managed the project. R.T. and I.S. wrote the original draft. All authors commented on and proofread the manuscript, with significant contributions from J.M.L., G.F., and S.L.

## Competing interests

The authors declare no competing interests.

## Additional information

¹Integrative Biology Laboratory, Nanjing Normal University, Nanjing 210023, China. ²BGI Research, Qingdao 266555, China. ³BGI Research, Shenzhen 518083, China. ⁴Qingdao Key Laboratory of Marine Genomics BGI Research, Qingdao 266555, China. ⁵Marine Mammal and Marine Bioacoustics Laboratory, Institute of Deep-sea Science and Engineering, Chinese Academy of Sciences, Sanya 572000, China. ⁶The Innovation Research Center for Aquatic Mammals, and Key

Laboratory of Aquatic Biodiversity and Conservation of the Chinese Academy of Sciences, Institute of Hydrobiology, Chinese Academy of Sciences, Wuhan 430072, China. [7]Key Laboratory of Animal Ecology and Conservation Biology, Institute of Zoology, Chinese Academy of Sciences, Beijing 100101, China. [8]School of Life Sciences, University of Science and Technology of China, Hefei 230027, China. [9]School of the Environment, The University of Queensland, Lucia 4072, Australia. [10]School of Veterinary Sciences, The University of Adelaide, Roseworthy 5371, Australia. [11]State Key Laboratory of Agricultural Genomics, BGI Research, Shenzhen 518083, China. [12]These authors contributed equally: Ran Tian, Yaolei Zhang, Hui Kang. ✉e-mail: fanguangyi@genomics.cn; lish@idsse.ac.cn; inge@seimlab.org

