## [Peer Review File · Nature Communications]

Sirenian genomes illuminate the evolution of fully aquatic species within the mammalian superorder AfrotheriaREVIEWER COMMENTS

Reviewer #1 (Remarks to the Author):

In this study the whole genomes of two dugongs have been sequenced and the chromosomal-level de novo assemblies were created. In addition, a transcriptome and whole genomes of 99 dugongs predominantly from the Eastern and Western coasts of Australia were generated. Phylogenetic and evolutionary analyses were performed by combining this data with the whole genome sequences of three dugongs (already published), manatees, and other Afrotherians from other studies. Furthermore, functional assays such as Iodine uptake were conducted. Finally, population genetic analyses were conducted using the 99 dugongs.

On the positive side, this study has created a large amount of data for a species that is located on a taxonomically unique branch (Sirenia), which is also a species considered vulnerable in terms of its conservation status. I don't see problems with the methodologies used for most of the analyses (but see below). Enough details are provided in the methods for reproduction of the results.

Other than the above, I didn't see anything new in the results presented here. Most of them were largely confirmatory. Some of the analyses were biased by picking up only a set of preconceived genes and not providing proper justification. I have provided a list of major concerns below.

1. First, what is the importance of generating two more de novo assemblies when there was a chromosomal-level assembly of dugong already available? Since there is no advantage (other than producing more data), the new assemblies do not contribute to the novelty of this study.
2. The authors searched for evolutionary signatures that suggested adaptation to the marine life of dugongs. For this purpose, they looked at the genes evolving under positive Darwinian selection, genes evolving rapidly, genes lost due to redundancy, and the Sirenian-specific substitutions in a selected set of genes. In my opinion, there was a serious flaw in all these analyses.
3. The authors describe four genes under positive selection, including ANPEP, ATP1B4, DUOXA2, and DIO1, and their role in surviving marine habitats. However, table S7 shows 30 genes under positive selection. Therefore, four out of 30 genes don't support any excess or enrichment of genes associated with marine adaptation. This is because, by analyzing the genome of any species, we will find a set of genes evolving under positive selection for various reasons. Unless a significant fraction of them is shown to be associated with adaptation to a specific habitat, it is difficult to be taken as evidence for that.
4. This bias was much more pronounced in the analysis of rapidly evolving genes (REGs). Table S8 shows 127 RAGs observed in Sirenia. However, the authors chose only four of them (ALB, KCNE2, SCN5A, and SERPINE2) and suggested their potential association with aquatic life. What about the rapid evolution of the remaining 123 genes? Almost all genomes will have a long list of fast-evolving genes, and we may not know the exact reason for those (assuming there were zero errors in identifying them). We need to first assume (as a null hypothesis) that the number of fast-evolving genes belonging to various functional categories or different habitat associations is equal. Of course, this expectation can be tested using a proper method to find out any excess or enrichment of genes belonging to a specific category (e.g., see Le Duc et al. 2022 Science Advances).
5. Similarly, the authors checked the Sirenian-specific amino acid changes ONLY in the genes involved in the thyroid hormone pathway and circadian clock genes (Table S11). It is well known that lineage-specific substitutions will occur in every gene purely by chance alone. The question to be tested here is whether the fraction of Sirenian-specific amino acid changes were significantly higher in the genes involved in thyroid hormone pathways and circadian clock genes than the rest of the genes (or the set of genes involved in various other pathways).
6. The same bias was observed in the analysis of the genes inactivated in Sirenia (Table S9). The authors selectively observed premature stop codons in only a small set of genes (chitinase, lysozyme, etc) that are presumed to be associated with marine life. A significant fraction of the remaining 7000 genes will also harbor such deletions and insertions. Was the fraction of indels in the former significantly higher than the latter?
7. Finally, many genes identified (e.g., ABCG8, ANPEP, CHIA5) to be associated with marine adaptation in this study have been reported previously in other marine mammalian species (as cited in the paper), and they are confirmatory.

Reviewer #2 (Remarks to the Author):

The manuscript by Ran et al. described the analysis on the genomes of dugong and related species to reveal genomic features and mechanisms for adaptations to aquatic species. Meanwhile, they also sequenced individuals from the dugong population to analyze the genomic characteristics of its population, reflecting their genetic diversity and demographic changes. The major findings and significant progresses include: 1) construction of a chromosome level genome of dugong, valuable for future related genomic studies; 2) identification of genes and genomic features underlying aquatic adaptations, as well as validation some of these features, improving our understanding on genetic mechanisms of aquatic mammal adaptation and evolution; 3) population genomics to reveal the population features and history, providing insights for future conservations. Overall, I think this study provides important biological insights, with valuable dataset and solid data analysis. However, I do have several major concerns regarding the proper presentation of the results and conclusions, as well as some minor points. I would suggest carefully revise the manuscript to clearly describe the methods and supporting information.

Major points

1. Although the focus of the main text would be the analysis on the genome sequence to reveal genomic features underpinning adaptations, critical information on the genome assembly and gene annotation should be provided. For example, in the first part of the result section (around Line 87-91), the authors should mention the number of the protein coding genes and in the method section, the authors should mention whether and how they used the RNA sequencing data for the annotation, and provide some supplementary information to indicate whether the gene annotation is of high quality (for example, comparing to related species to show whether their gene number is comparable). This is just one example, which I think the authors should revise the manuscript. Other than critical information on genome assembly and annotation, important information on how the authors identify critical genome features (gene loss, positively selected genes, etc.) should also be briefly mentioned in the main text.
2. In addition to the first point, I also found the manuscript to be quite long and, in some cases, distracting. I understand that the dataset generated here can be analyzed in different aspects, but only critical analysis and results related should be included. For example, in Line 515-516, the author mentioned k-mer analysis, but I did not find any place the authors mentioned the results. The authors should carefully revise the manuscript to make it brief and concise, especially for methods and supporting information. Another example, there are more than one hundred references included, which I think are too many.
3. The current manuscript contains two parts, including the first part to investigate the dugong genome, and the second part to analyze the dugong population. I think these two parts seem to be quite separated, which might be modified, especially considering that there are genomic features identified in both parts. For example, is it possible to investigate the genes identified through the genome analysis (Figure 2) in the population dataset to see whether these genes were conserved or not in the population? Similarly, for the CLPX and other genes identified in the population analysis, how about their homologous genes or sequences in other related species? Although population level evolution and the resulting genome features should be more recent than the species level evolution and the resulting genome features, I would anticipate the two sub-datasets to support each other thus more solid conclusions can be made.

Minor points

Line 87, full name of stLFR should be mentioned here.

Line 280, it is not very clear by saying 'persistent organic pollutants'

Line 275-287, this paragraph can be shortened.

Line 320, is it true cetaceans lost first nine exons? Results or references should be provided here.

Line 328-331, it is not clear here how the authors get to this. Maybe, references should be provided.

Line 343-346, this sentence seems to be problematic. You might want to revise it to indicate you obtain individuals from two other locations and sequenced them.

Line 350-353, do you mean the exactly same individual?

Line 355, you should mention effective population size here instead of directly N_e .

Line 356, by saying 'All dugongs', do you mean representative individuals' data to reflect the dugong population change? Although PSMC can be applied to single individual's variation dataset to infer the population change, multiple individuals' result also just indicate the overall population change instead of multiple sub-population changes. The results from difference individuals can be consistent. I think you should revise this sentence as well as several later sentences to clearly indicate that and avoid misleading.

Line 381-398, the authors identified a region under positive selection in Northern population. Positive selection should reflect alterations of genotypes and to have been adapted through these alterations. If so, the Northern population should have specific genotypes for their adaption to Northern Queensland environmental conditions. I would suggest the authors to revise these sentences to better indicate what the positive selection might have reflected.

Line 399-409, sometimes, the genomic diversity cannot directly reflect the extinction events (in some cases, it was called as genetic diversity extinction debt). So the descriptions and discussions in this paragraph should be revised here.

Line 476, please confirm whether the word 'cow' to be right here.

Line 488, full name for DSMO should be provided here.

Line 499, full coma missing here.

Line 511-513, how the RNA was extracted should be provided here. I would also suggest comprehensively look into the online method section to clearly indicate methods used in different part. The RNA sequencing part is one example of not so clear method description.

Line 538-540, 'de novo' should be italic. And the protein coding gene annotation was repeatedly mentioned here and in Line 549-551.

Line 666, why the authors used the human protein for the alignment and identification of gene loss events instead of using other more closely related species?

Line 668-669, duplicated 'mapped' here.

Line 837, the writing of F_{st} is not right here.

Line 1505-1506, the sentence seems to be not complete.

RESPONSE TO REVIEWERS' COMMENTS

We thank the reviewers for their careful reading, helpful comments, and constructive suggestions that have significantly improved our manuscript. Our point-by-point responses are detailed below. Given our extensive edit, please note that a tracked version of our manuscript Word document is provided as a Related Manuscript File PDF.

Reviewer #1 (Remarks to the Author):

In this study the whole genomes of two dugongs have been sequenced and the chromosomal-level de novo assemblies were created. In addition, a transcriptome and whole genomes of 99 dugongs predominantly from the Eastern and Western coasts of Australia were generated. Phylogenetic and evolutionary analyses were performed by combining this data with the whole genome sequences of three dugongs (already published), manatees, and other Afrotherians from other studies. Furthermore, functional assays such as Iodine uptake were conducted. Finally, population genetic analyses were conducted using the 99 dugongs.

On the positive side, this study has created a large amount of data for a species that is located on a taxonomically unique branch (Sirenia), which is also a species considered vulnerable in terms of its conservation status. I don't see problems with the methodologies used for most of the analyses (but see below). Enough details are provided in the methods for reproduction of the results.

Other than the above, I didn't see anything new in the results presented here. Most of them were largely confirmatory. Some of the analyses were biased by picking up only

a set of preconceived genes and not providing proper justification. I have provided a list of major concerns below.

We agree that some previous studies revealed a partial genetic basis for aquatic adaptations by sirenians (like the skin). However, we have some innovations in our article: 1) we are the first to show changes in sirenian thyroid metabolism and the circadian clock genes, and we performed a cell line experiment to test the function of selected genes. 2) we resequenced 99 dugongs and provide new population genomics insights – including a candidate ecotype on the Australian east coast. Please also see our responses below.

Comment 1.1: *First, what is the importance of generating two more de novo assemblies when there was a chromosomal-level assembly of dugong already available? Since there is no advantage (other than producing more data), the new assemblies do not contribute to the novelty of this study.*

Response

The novelty of our manuscript lies in generating and annotating a dugong genome suitable to carry out comparative genomics to reveal candidate genomic changes associated with sirenian adaptations and the population genomics of ~100 dugong individuals. However, we agree that this point should be made more explicit in the manuscript text and have now attempted to do so.

To expand, when we commenced the work on the manuscript (late 2021), no high-quality dugong genome assembly was available. The only assembly at that time was poor quality and assembled from short-insert libraries. The later assemblies became publicly available in January 2022 (DNA Zoo), February 2022 (Max-Planck Institute for Evolutionary Anthropology ¹), and May 2023 (Vertebrate Genome Project ²) (see **Supplementary Note 1**). Although the genome quality in this study is comparable with the previously reported genomes, we agree that our new dugong assembly was perhaps not novel at the time of manuscript submission (August 2023). However, our

reference genome was necessary [and of suitable quality] for us to initiate this project to do comparative and population genomics analyses in early 2022.

Comment 1.2: The authors searched for evolutionary signatures that suggested adaptation to the marine life of dugongs. For this purpose, they looked at the genes evolving under positive Darwinian selection, genes evolving rapidly, genes lost due to redundancy, and the Sirenian-specific substitutions in a selected set of genes. In my opinion, there was a serious flaw in all these analyses.

Response

Thank you for raising this critical point. We agree that the way our genes were presented in our submission makes them appear ‘cherry-picked’ and apologize to both reviewers for omitting a most critical justification in our introductory Results section. We now outline the logic of our approach to identifying genes of interest – a strategy similar to other recent manuscripts (e.g., see ^{1,3,4}). That is, [gene] selection and literature searches were first used to identify genes that may be associated with sirenian-specific features, and genes that may have facilitated their transition to aquatic life (see specific responses below).

The Result section now includes the following text that refers to a new subsection in the Methods that details our analysis strategy: “Our afrotherian dataset, which included the West Indian manatee and the phylogenetically closest extant terrestrial species to sirenians (elephants and hyraxes) (**Supplementary Note 2, Figures S1c and S3**), was interrogated (see ‘Comparative genomics analysis strategy’ in Methods and **Tables S5-S13**) to illuminate features present since the sirenian crown ancestor (**Figure 1**) that may underlie aquatic herbivory, sirenian circadian activity patterns, and typical marine mammal features such as modified cardiovascular (**Supplementary Note 3**), integumental (i.e., skin and associated structures), and sensory (vision, smell, and taste) systems”.

“**Comparative genomics analysis strategy.** To summarize our analysis strategy (see

detailed methods below) and manuscript data, we first compared signatures of natural selection with literature searches (comprehensive reviews on the anatomical and physiological adaptations of sirenians to aquatic life, including references ^{5,6}) to discover broad functional categories associated with sirenian adaptations. Enrichment analysis of positively selected (**Tables S5 and S6**) and rapidly evolving (**Tables S7 and S8**) genes using KOBAS revealed an over-representation (Benjamini–Hochberg $P < 0.05$) of terms related to thyroid hormone synthesis, the cardiovascular system, integumentary system (i.e., cornified envelope), and circadian activity. Sirenian-specific amino acid substitutions in the thyroid hormone pathway and circadian clock proteins were next identified using FasParser ^{7,8} (**Table S9**) and validated against our genome resequencing data set of 99 dugongs (see later), the 120 mammalian species in OrthoMaM ⁹, and by BLAST ¹⁰ searches of NCBI and Ensembl databases. Functional *in vitro* assays were used to evaluate selected substitutions. CAFE ¹¹ revealed loss of gene families of the integumentary (i.e., cornification/keratinization) and olfactory systems (**Tables S10 and S11**). A recently described pipeline ¹² confirmed reported pseudogenes (e.g., refs. ¹³ and ¹⁴) among the 15 shared by sirenians (**Tables S12 and S13**) but also gene inactivation events not previously described”.

***Comment 1.3:** The authors describe four genes under positive selection, including ANPEP, ATP1B4, DUOXA2, and DIO1, and their role in surviving marine habitats. However, table S7 shows 30 genes under positive selection. Therefore, four out of 30 genes don't support any excess or enrichment of genes associated with marine adaptation. This is because, by analyzing the genome of any species, we will find a set of genes evolving under positive selection for various reasons. Unless a significant fraction of them is shown to be associated with adaptation to a specific habitat, it is difficult to be taken as evidence for that.*

Response

We agree and apologize for not clearly outlining how we associated PSGs and REGs (see response to **Comment 1.4**) with particular functions/candidate adaptations. As outlined above, we combined enrichment analysis of genes under natural selection (in

the new **Table S6** for PSGs and **Table S8** for REGs) and literature searches to identify adaptations of interest – similar to Le Duc and colleagues’ manuscript ¹ (e.g., see Table S8 legend in their supplementary information).

As we mentioned above, we used KOBAS to perform enrichment analysis (employs a hypergeometric/Fisher’s exact test for statistical test and Benjamini and Hochberg for multiple testing correction (see **tables S6 and S8**). KOBAS is similar to the FUCN package used by Le Duc *et al.* 2022.

Comment 1.4: *This bias was much more pronounced in the analysis of rapidly evolving genes (REGs). Table S8 shows 127 RAGs observed in Sirenia. However, the authors chose only four of them (ALB, KCNE2, SCN5A, and SERPINE2) and suggested their potential association with aquatic life. What about the rapid evolution of the remaining 123 genes? Almost all genomes will have a long list of fast-evolving genes, and we may not know the exact reason for those (assuming there were zero errors in identifying them). We need to first assume (as a null hypothesis) that the number of fast-evolving genes belonging to various functional categories or different habitat associations is equal. Of course, this expectation can be tested using a proper method to find out any excess or enrichment of genes belonging to a specific category (e.g., see Le Duc et al. 2022 Science Advances).*

Response

See response to **Comment 1.3**.

Comment 1.5: *Similarly, the authors checked the Sirenian-specific amino acid changes ONLY in the genes involved in the thyroid hormone pathway and circadian clock genes (Table S11). It is well known that lineage-specific substitutions will occur in every gene purely by chance alone.*

Response

We detected about 4,000 genes with unique AA of sirenians in our Afrotheria dataset (eight afrotherian species and sloth as outgroup). Obviously, with such a small number of species and ~7,000 orthologs, gene enrichment analysis of these genes is

not informative. However, considering the selection signals and enrichment of genes related to the thyroid hormone pathway and the circadian clock, we further focused on sirenian-specific amino acid changes of genes in these pathways (including manual validation of raw sequencing reads). We now clearly state that the unique amino acids of these genes are shown after 1. gene enrichment analysis of natural selection data (PSG and REGs – see responses to **Comment 1.3** and **1.4**) and 2. literature searches (see response to **Comment 1.2**).

***Comment 1.6:** The same bias was observed in the analysis of the genes inactivated in Sirenia (Table S9). The authors selectively observed premature stop codons in only a small set of genes (chitinase, lysozyme, etc) that are presumed to be associated with marine life. A significant fraction of the remaining 7000 genes will also harbor such deletions and insertions. Was the fraction of indels in the former significantly higher than the latter?*

Response

Gene loss events (pseudogenes) were identified using a recently described pipeline¹² and combined with manual validation of raw genome and transcriptome reads using BLAST¹⁰.

Briefly, the gene losses in each species are defined as genes harboring disrupting mutations, including premature stop codons and/or frameshifts with intact 1:1 orthologs in our dataset using human as the reference. Although this pipeline does not consider the fraction of indels, it sets strict screening standards to ensure the inactivation sites are correct. We manually validated all candidate pseudogenes. A total of 15 sirenian pseudogenes were identified (**Table S12**). No genes were significantly enriched in KOBAS (data not shown). Gene enrichment using STRING revealed enrichment (in new **Table S13**) for a single manuscript on cetacean skin genes with inactivating mutations (4/15 genes. Please note that STRING v12.0 only includes manuscripts up to mid-2022¹⁵, thus some recent manuscripts on cetacean skin-associated gene loss were not interrogated by the STRING tool). Nevertheless,

the enrichment of a single manuscript further supports that most lost genes are related to the skin and its appendages.

For clarity, we have added the following to the ‘The integumentary system’ Results section: “We identified and validated using dugong epidermis RNA-seq reads the loss of multiple skin-associated genes (**Figure 3b** and **Table S12**). Notably, many of these genes are convergently lost in cetaceans, as revealed by manual literature searches and STRING¹⁵ gene enrichment of the 15 shared sirenian pseudogenes (**Table S13** and **Supplementary Note 5**)”.

***Comment 1.7:** Finally, many genes identified (e.g., ABCG8, ANPEP, CHIA5) to be associated with marine adaptation in this study have been reported previously in other marine mammalian species (as cited in the paper), and they are confirmatory.*

Response

We include these to indicate that our gene pseudogene pipeline, which revealed a total of 15 genes, can identify novel gene inactivation events as well as the loss of various previously reported genes (including *PONI*¹³) and to put their loss into a broader functional context. Please note that we have added the discussion of these genes to the new **Supplementary Note 6**.

To clearly make the above point, see the final sentence of the ‘Comparative genomics analysis strategy’: “A recently described pipeline¹² confirmed reported pseudogenes (e.g., refs.¹³ and¹⁴) among the 15 shared by sirenians (**Tables S12** and **S13**) but also gene inactivation events not previously described”.

We have also moved gene loss results, that are confirmatory to other studies, to various Supplementary Notes (Supplementary Note 4. Sirenian herbivory; Supplementary Note 5. Molecular evolution of the sirenian integumentary system; and Supplementary Note 6. Convergent loss of *PONI* and *CES3*).

Reviewer #2 (Remarks to the Author):

The manuscript by Ran et al. described the analysis on the genomes of dugong and related species to reveal genomic features and mechanisms for adaptations to aquatic species. Meanwhile, they also sequenced individuals from the dugong population to analyze the genomic characteristics of its population, reflecting their genetic diversity and demographic changes. The major findings and significant progresses include: 1) construction of a chromosome level genome of dugong, valuable for future related genomic studies; 2) identification of genes and genomic features underlying aquatic adaptations, as well as validation some of these features, improving our understanding on genetic mechanisms of aquatic mammal adaptation and evolution; 3) population genomics to reveal the population features and history, providing insights for future conservations. Overall, I think this study provides important biological insights, with valuable dataset and solid data analysis. However, I do have several major concerns regarding the proper presentation of the results and conclusions, as well as some minor points. I would suggest carefully revise the manuscript to clearly describe the methods and supporting information..

Comment 2.1: *Although the focus of the main text would be the analysis on the genome sequence to reveal genomic features underpinning adaptations, critical information on the genome assembly and gene annotation should be provided. For example, in the first part of the result section (around Line 87-91), the authors should mention the number of the protein coding genes and in the method section, the authors should mention whether and how they used the RNA sequencing data for the annotation, and provide some supplementary information to indicate whether the gene annotation is of high quality (for example, comparing to related species to show whether their gene number is comparable). This is just one example, which I think the authors should revise the manuscript. Other than critical information on genome assembly and annotation, important information on how the authors identify critical genome features (gene loss, positively selected genes, etc.) should also be briefly mentioned in the main text.*

Response

We agree and have now added a new Results section titled ‘An annotated dugong genome for comparative and population genomic analyses’.

We have added the number of annotated protein-coding genes (18,663) in our assembly (19,897 predicted) to the Results section (see **Table S1**). Considering that different gene annotation pipelines can produce different gene numbers (e.g., contrast any genome assembly available in NCBI and Ensembl), directly comparing gene numbers can be difficult. However, our predicted gene number is similar to that of the afrotherian NCBI assemblies in our dataset, while the DNA Zoo gene annotation appears to predict a larger number of genes: NCBI (Cape golden mole; 19,530; Lesser hedgehog tenrec, 19,805; Cape elephant shrew, 20,207) vs. DNA Zoo (dugong, 25,074; manatee, 25,094; rock hyrax, 22,880; Asian elephant 26,246). As indicated by **Table S3** (and also see **Supplementary Note 1**), BUSCO analysis also indicates that our assembly and annotations are comparable to the other species in our dataset.

The Online Methods section (‘Genome annotation’) details how RNA-sequencing data were used: “Transcriptome data (clean reads) were mapped to the assembled genome using HISAT2 v2.1.0¹⁶ and SAMtools v1.9¹⁷, and coding regions were predicted using TransDecoder v5.5.0^{18,19}”.

As outlined in our response to **Comment 1.2** above, we also now outline the overall strategy employed to identify gene changes that potentially play a role in sirenian adaptations.

Comment 2.2: *In addition to the first point, I also found the manuscript to be quite long and, in some cases, distracting. I understand that the dataset generated here can be analyzed in different aspects, but only critical analysis and results related should be included. For example, in Line 515-516, the author mentioned k-mer analysis, but I did not find any place the authors mentioned the results. The authors should carefully revise the manuscript to make it brief and concise, especially for methods and supporting*

information. Another example, there are more than one hundred references included, which I think are too many.

Response

We agree and have removed the *k*-mer sentence from the Online Methods.

Given that *Nature Communications* is online-only and does not have a reference limit, we opted to include a thorough reference list rather than adding references to support selected arguments (e.g., see recent manuscript by one of the co-authors of this manuscript²⁰). We have, nevertheless, attempted to reduce the number of references in the main manuscript. The number of references in the *Main* text is now 88, down from 120. The manuscript text (up to Online Methods) is now 4,074 words, down from ~5,078. The main text should now better outline the novel findings of our study, while the Supplementary Notes include confirmatory data (for example, on diet and integumentary system gene loss) or data without major data items (figures or tables) in the *Main* text beyond summary **Figure 2** (here, the sirenian cardiovascular system).

Specifically, we attempted to streamline our manuscript by moving some results to the supplementary information:

- Supplementary Note 3. Molecular evolution of the sirenian cardiovascular system’.
- ‘Supplementary Note 4. Sirenian herbivory’
- ‘Supplementary Note 6. Convergent loss of *PONI* and *CES3*’.

The associated aquatic herbivory Results section now states:

“Sirenians are the only aquatic herbivorous mammals, and we observed gene losses consistent with a diet comprising few animal products (**Table S12** and **Supplementary Note 4**).

...

“We identified three gene activation events that may be disadvantageous today:

convergent loss of *PONI* and *CES3* in marine mammals (**Supplementary Note 6** and **Figures S5** and **S6**) and sirenian-specific loss of *KCNK18*".

We also moved the following text to **Supplementary Note 4**: "A study of captive West Indian manatees found that switching their diet from terrestrial plants to seagrass over 19 days increased blood thyroid hormone levels ²¹ – as expected given the difference in dietary iodine content. Wild manatees showed higher levels of circulating thyroid hormones than any captive diet group ²¹, which we hypothesize resulted from a lifetime on a high-iodine seagrass diet and associated energy metabolism balance."

***Comment 2.3:** The current manuscript contains two parts, including the first part to investigate the dugong genome, and the second part to analyze the dugong population. I think these two parts seem to be quite separated, which might be modified, especially considering that there are genomic features identified in both parts. For example, is it possible to investigate the genes identified through the genome analysis (Figure 2) in the population dataset to see whether these genes were conserved or not in the population? Similarly, for the CLPX and other genes identified in the population analysis, how about their 6 of 8 homologous genes or sequences in other related species? Although population level evolution and the resulting genome features should be more recent than the species level evolution and the resulting genome features, I would anticipate the two sub- datasets to support each other thus more solid conclusions can be made.*

Response

As outlined by the reviewer, we expect the likely drivers of shared sirenian traits (adaptations to a fully aquatic life by the dugong and West Indian manatee more than 30 million years ago) to not directly overlap with drivers of an apparent recent genetic barrier in dugongs on the Queensland coast. However, we do understand (and appreciate) that a logical question is to ask whether there is indeed an overlap of genetic changes.

As a consequence of this reviewer's suggestion, we have now examined the genes under positive selection in the northern Queensland dugong group further. Please note that after manual curation of the gene loci, we determined *CLPX-like* to be a processed dugong-specific pseudogene (no afrotherian orthologs and numerous inactivating mutations) and have removed it from **Table S16**, leaving five genes under selection in the comparison of the southern and northern Queensland dugongs. In afrotherians (and other mammals) we detected orthologs of the non-immunoglobulin genes, *NUP42* and *CLPX*.

- We found no evidence of positive or rapid selection (PSGs and REGs) of these genes in sirenians (West Indian manatee and dugong; **Tables S5** and **S7**) or the dugong alone (our assembly, Ddugon_BGI, was generated from a southern Queensland group individual). Please see **Tables R1** and **R2** below for PSG and REG analysis statistics.
- A *CLPX* SNP/genotype results in an acid change (Ile197Thr) in northern Queensland dugongs unique to this population group – compared to the southern Queensland dugongs, other sirenians (i.e., West Indian manatee and Steller's sea cow), and the 120 mammalian species in OrthoMaM. After consulting the literature and internet resources, we cannot currently propose a functional effect of this substitution, however (predicted as 'benign' by PolyPhen-2 and 'tolerated' by SIFT; **Table S15**). It does, however, hint at a candidate ecotype.
- The 24 genes in **Table S9** are sirenian-specific – that is, found in our assembly and 99 other Australian dugongs, the West Indian manatee, and Steller's sea cow. In the Methods section 'Comparative genomics analysis strategy', we now state: "Sirenian-specific amino acid substitutions in the thyroid hormone pathway and circadian clock proteins were next identified using FasParser^{7,8} (**Table S9**) and validated against our genome resequencing data set of 99 dugongs (see later), the 120 mammalian species in OrthoMaM⁹, and by BLAST¹⁰ searches of NCBI and Ensembl databases".

Table R1 | PAML branch-site test of selection in sirenian (dugong and West Indian manatee) and the dugong. Note that the dugong examined corresponds to assembly Ddugon_BGI (an individual from Hervey Bay, the southern Queensland coast group in our analysis).

Gene	Branch	Model	lnL	2ΔlnL	P-value	Parameters
CLPX	Ancestor of sirenians	Ma0	-4897.84			$\omega_0=0.03383,$ $\omega_2=1.00000$
	Ancestor of sirenians	Ma	-4897.84	0	1	$\omega_0=0.03383,$ $\omega_2=1.00000$
NUP42	Ancestor of sirenians	Ma0	-5223.67			$\omega_0=0.13475,$ $\omega_2=1.00000$
	Ancestor of sirenians	Ma	-5223.67	0	1	$\omega_0=0.13475,$ $\omega_2=1.00000$
CLPX	Branch of dugong	Ma0	-4897.84			$\omega_0=0.03383,$ $\omega_1=1.00000, \omega_2=1.00000$
	Branch of dugong	Ma	-4897.84	0	1	$\omega_0=0.03383,$ $\omega_1=1.00000, \omega_2=1.00000$
NUP42	Branch of dugong	Ma0	-5135.03			$\omega_0=0.14110,$ $\omega_1=1.00000, \omega_2=1.00000$
	Branch of dugong	Ma	-5135.03	0	1	$\omega_0=0.14110,$ $\omega_1=1.00000, \omega_2=1.00000$

Table R2 | PAML branch test of selection in sirenian (dugong and West Indian manatee) and the dugong. Note that the dugong examined corresponds to assembly Ddugon_BGI (an individual from Hervey Bay, the southern Queensland coast group in our analysis).

Gene	Branch	Model	lnL	2ΔlnL	P-value	Parameters
CLPX	Ancestor of sirenians	One_ratio	-4904.82			$\omega_0=0.05055$
	Ancestor of sirenians	Two_ratio	-4904.74	0.149324	0.699182207	$\omega_0=0.05112; \omega_1=0.04030$
NUP42	Ancestor of sirenians	One_ratio	-5269.57			$\omega_0=0.39606$
	Ancestor of sirenians	Two_ratio	-5269.45	0.243908	0.62139769	$\omega_0=0.39955; \omega_1=0.33142$
CLPX	Branch of dugong	One_ratio	-4904.82			$\omega_0=0.05055$
	Branch of dugong	Two_ratio	-4904.81	0.002694	0.958605381	$\omega_0=0.05051; \omega_1=0.05354$
NUP42	Branch of dugong	One_ratio	-5269.57			$\omega_0=0.39606$
	Branch of dugong	Two_ratio	-5269.51	0.120124	0.72890004	$\omega_0=0.39755; \omega_1=0.32917$

Minor points

Line 87, full name of stLFR should be mentioned here.

Done.

Line 280, it is not very clear by saying 'persistent organic pollutants'

We agree and have added a description in brackets: “organic pollutants (i.e., organic chemicals that persist in the environment)”. Note that this text has now been moved to **Supplementary Note 6**.

Line 275-267, this paragraph can be shortened.

Done. In particular, we simplified to final sentence from “Taken together, the lack of a pineal gland and a genetic background where many circadian genes have unique amino acid changes or are lost strengthens the idea that the sirenian circadian clock has been recalibrated”

to (now in the Discussion, in an effort to better put the circadian system results in context):

“The lack of a pineal gland and their genetic background support that the circadian clock (i.e., sleep-wake cycle) of sirenians has been recalibrated (e.g., see ²²), likely to facilitate an activity pattern in a more light-limited, fully aquatic environment heavily reliant on lunar tidal currents and water temperature fluctuations”.

Line 320, is it true cetaceans lost first nine exons? Results or references should be provided here.

This is correct. In addition to several inactivating mutations, we could not detect the first nine exons in cetaceans.

As indicated in the following paragraph (now in **Supplementary Note 6**), our manual validation of the pseudogene loss confirmed previously cited reports on this gene inactivation: “We identified loss of carboxylesterase 3 (*CES3*; also known as *ES31*) in sirenians (**Table S12**), cetaceans (loss of first nine coding exons and downstream

inactivating mutations), and phocids (inactivating mutations, including a 10-bp deletion in Phocidae, the largest pinniped family²³) (**Figure S6**). Loss of *CES3* – by the West Indian manatee and killer whale¹⁴, and by four cetaceans and two hippos²⁴ – has previously been reported but not discussed. A premature stop codon is shared by all sirenians, while the dugong and Steller’s sea cow share an additional stop codon (**Figure S6**).

Line 328-331, it is not clear here how the authors get to this. Maybe, references should be provided.

For clarity, we now cite the references from the preceding sentence again in **Supplementary Note 6**: “While *CES3* expression is much lower than *CES1* and its enzyme has several magnitudes lower catalytic efficiency than *CES1* for many compounds²⁵⁻²⁷ (and, thus, *CES3* loss in marine mammals is likely compensated), carboxylesterase 3 may show exclusive specificity against manufactured compounds such as pesticides”.

Line 343-346, this sentence seems to be problematic. You might want to revise it to indicate you obtain individuals from two other locations and sequenced them.

To clarify that we only generated resequencing data of the 99 Queensland individuals, we have reworded the section to read: “Here, we considered the population genomics of dugongs from ten locations (**Figure 5a**). To this end, we generated short-read whole-genome resequencing data from seven locations (99 individuals) spanning 2,000 km of the Australian east coast (from Torres Strait to Moreton Bay, Queensland) (**Table S14**). We obtained 3.46 Tb of data, with an average sequencing depth of 11.41×, and identified 71.25 million high-quality SNPs (average SNP density 24.61 SNPs/kb). Publicly available resequencing data (one individual carcass each) was also obtained from two other Australian locations, Coogee Beach (New South Wales)¹ and Exmouth Gulf (Western Australia), and from waters off Okinawa (Japan)”.

Line 350-353, do you mean the exactly same individual?

No. We have changed the wording to read: “Because this individual stranded ~750 km from the accepted eastern Australian range during the summer (November), we propose it represents one of the few instances ²⁸ of seasonal long-distance ranging from a population in close geographic proximity to Moreton Bay”.

Line 355, you should mention effective population size here instead of directly N_e .

Done. The text now reads: “...to track changes in effective population size (N_e ; the number of individuals that will contribute to the next population) during the Pleistocene...”.

Line 356, by saying ‘All dugongs’, do you mean representative individuals’ data to reflect the dugong population change? Although PSMC can be applied to single individual’s variation dataset to infer the population change, multiple individuals’ result also just indicate the overall population change instead of multiple sub-population changes. The results from difference individuals can be consistent. I think you should revise this sentence as well as several later sentences to clearly indicate that and avoid misleading.

We agree that that the term “All dugongs” is misleading here. We have reworded it to “all examined dugongs”.

Line 381-398, the authors identified a region under positive selection in Northern population. Positive selection should reflect alterations of genotypes and to have been adapted through these alterations. If so, the Northern population should have specific genotypes for their adaption to Northern Queensland environmental conditions. I would suggest the authors to revise these sentences to better indicate what the positive selection might have reflected.

We thank the reviewer for pointing this missing (and critical) analysis out. We have now examined whether the detected selection acting on the northern population result in SNP genotypes that alter amino acid residues (see response to **Comment 2.3**).

In the Discussion, we also further stress that more work is needed on potential environment-mediated selection: “We also confirm ²⁹ and (for the first time) date a north-south genetic break that emerged approximately 10.7 thousand years ago on the Australian east coast and reveal a ~2 Mb genetic sweep region that may be associated with historical and recurrent environmental differences between the north and south coast and formation of an ecologically distinct population (ecotype ³⁰). Our dataset allows future explorations of genetic structure related to geographical region and environmental variables”.

Line 399-409, sometimes, the genomic diversity cannot directly reflect the extinction events (in some cases, it was called as genetic diversity extinction debt). So the descriptions and discussions in this paragraph should be revised here.

We thank the reviewer for pointing this very important issue out.

We agree that stating that “extinction events can often be predicted from the genetic history of a species, with loss of genomic diversity reflecting a dwindling population” is inaccurate. Reduced genetic diversity, as recent genome-wide studies show (e.g., on the vaquita ³¹⁻³³), may indeed not always reflect extinction events or the risk of extinction. For example, genome-wide heterozygosity should be considered in concert with functional genetic diversity metrics such as deleterious and loss-of-function alleles (LOFs) ^{31,32}.

The lower heterozygosity of the Okinawan dugong is now briefly mentioned later in the manuscript: “The historical demography (PSMC) of the Okinawan individual aligned with its genome diversity estimates. It had one magnitude of order lower genome-wide heterozygosity (5.65×10^{-4}) compared with Australian dugongs ($\sim 1 \times 10^{-3}$). One-third of its genome was in ROH segments above one megabase (389 ROHs spanning 934.4 Mb), with evidence of inbreeding as recently as 135 years ago ($F_{ROH>10Mb} = 0.025$, four ROHs spanning 73.6 Mb) to 54 years ago ($F_{ROH>20Mb} = 0.010$, one ROH spanning 29.2 Mb)”.

Line 476, please confirm whether the word 'cow' to be right here.

Correct. A term used to describe an adult female sirenian.

Line 488, full name for DSMO should be provided here.

Done.

Line 499, full coma missing here.

Fixed.

Line 511-513, how the RNA was extracted should be provided here. I would also suggest comprehensively look into the online method section to clearly indicate methods used in different part. The RNA sequencing part is one example of not so clear method description.

We now state the following in the Methods sections: “RNA from fetal liver (sample D201106) and skin (sample D110419), extracted using an RNeasy Mini Kit (QIAGEN), was sequenced on the BGISEQ-500 platform to generate 86.7 and 96.3 Gb of 150-bp paired-end read RNA-seq data, respectively”.

Line 538-540, 'de novo' should be italic. And the protein coding gene annotation was repeatedly mentioned here and in Line 549-551.

Fixed.

Line 666, why the authors used the human protein for the alignment and identification of gene loss events instead of using other more closely related species?

We employed the human as the reference species in the pipeline since it has a high-quality, well-annotated assembly – a requirement of the pseudogene detection pipeline employed¹². Given the extensive associated literature and database resources of human genes, it is also an ideal surrogate to infer function. After manually validating pseudogenes, we identified 15 sirenian pseudogenes (i.e., shared by crown Sirenia). It

is appreciated, however, that afrotherian-specific genes would be missed using this reference species.

Line 668-669, duplicated 'mapped' here.

Fixed.

Line 837, the writing of Fst is not right here.

Fixed (further clarified): "... Next, the pairwise fixation index (*Fst*) was calculated between the seven Queensland locations and between the whole northern (Torres Strait, Bowling Green Bay, and Airlie Beach) and southern (Moreton Bay, Great Sandy Straits, Hervey Bay, and Clairview) groups from Queensland...".

Line 1505-1506, the sentence seems to be not complete.

Fixed (added "in sirenians").

References

1. Le Duc, D. *et al.* Genomic basis for skin phenotype and cold adaptation in the extinct Steller's sea cow. *Sci Adv* **8**, eabl6496 (2022).
2. Baker, D.N. *et al.* A chromosome-level genome assembly for the dugong (*Dugong dugon*). *J Hered* (2024).
3. Yuan, J. *et al.* How genomic insights into the evolutionary history of clouded leopards inform their conservation. *Sci Adv* **9**, eadh9143 (2023).
4. Cole, T.L. *et al.* Genomic insights into the secondary aquatic transition of penguins. *Nat Commun* **13**, 3912 (2022).
5. Berta, A., Sumich, J.L. & Kovacs, K.M. *Marine Mammals*, 1-14 (Elsevier, 2015).
6. Marsh, H., O'Shea, T.J. & Reynolds III, J.E. *Ecology and conservation of the Sirenia: dugongs and manatees*, (Cambridge University Press, 2012).
7. Sun, Y.B. FasParser2: a graphical platform for batch manipulation of tremendous amount of sequence data. *Bioinformatics* **34**, 2493-2495 (2018).
8. Sun, Y.B. FasParser: a package for manipulating sequence data. *Zool Res* **38**, 110-112 (2017).
9. Scornavacca, C. *et al.* OrthoMaM v10: Scaling-Up Orthologous Coding Sequence and Exon Alignments with More than One Hundred Mammalian Genomes. *Mol Biol Evol* **36**, 861-862 (2019).
10. Camacho, C. *et al.* BLAST+: architecture and applications. *BMC*

- Bioinformatics* **10**, 421 (2009).
11. De Bie, T., Cristianini, N., Demuth, J.P. & Hahn, M.W. CAFE: a computational tool for the study of gene family evolution. *Bioinformatics* **22**, 1269-71 (2006).
 12. Zheng, Z., Hua, R., Xu, G., Yang, H. & Shi, P. Gene losses may contribute to subterranean adaptations in naked mole-rat and blind mole-rat. *BMC Biol* **20**, 44 (2022).
 13. Meyer, W.K. *et al.* Ancient convergent losses of Paraoxonase 1 yield potential risks for modern marine mammals. *Science* **361**, 591-594 (2018).
 14. Huelsmann, M. *et al.* Genes lost during the transition from land to water in cetaceans highlight genomic changes associated with aquatic adaptations. *Sci Adv* **5**, eaaw6671 (2019).
 15. Szklarczyk, D. *et al.* The STRING database in 2023: protein-protein association networks and functional enrichment analyses for any sequenced genome of interest. *Nucleic Acids Res* **51**, D638-D646 (2023).
 16. Kim, D., Paggi, J.M., Park, C., Bennett, C. & Salzberg, S.L. Graph-based genome alignment and genotyping with HISAT2 and HISAT-genotype. *Nat Biotechnol* **37**, 907-915 (2019).
 17. Li, H. *et al.* The Sequence Alignment/Map format and SAMtools. *Bioinformatics* **25**, 2078-9 (2009).
 18. Grabherr, M.G. *et al.* Full-length transcriptome assembly from RNA-Seq data without a reference genome. *Nat Biotechnol* **29**, 644-52 (2011).
 19. Haas, B.J. *et al.* De novo transcript sequence reconstruction from RNA-seq using the Trinity platform for reference generation and analysis. *Nat Protoc* **8**, 1494-512 (2013).
 20. Zhu, P. *et al.* Correlated evolution of social organization and lifespan in mammals. *Nat Commun* **14**, 372 (2023).
 21. Ortiz, R.M., Mackenzie, D.S. & Worthy, G.A. Thyroid hormone concentrations in captive and free-ranging West Indian manatees (*Trichechus manatus*). *Journal of Experimental Biology* **203**, 3631-3637 (2000).
 22. Mukhametov, L.M., Lyamin, O.I., Chetyrbok, I.S., Vassilyev, A.A. & Diaz, R.P. Sleep in an Amazonian manatee, *Trichechus inunguis*. *Experientia* **48**, 417-9 (1992).
 23. Paterson, R.S., Rybczynski, N., Kohno, N. & Maddin, H.C. A total evidence phylogenetic analysis of pinniped phylogeny and the possibility of parallel evolution within a monophyletic framework. *Frontiers in Ecology and Evolution* **7**, 457 (2020).
 24. Springer, M.S. *et al.* Genomic and anatomical comparisons of skin support independent adaptation to life in water by cetaceans and hippos. *Curr Biol* **31**, 2124-2139 e3 (2021).
 25. Zhao, B., Bie, J., Wang, J., Marqueen, S.A. & Ghosh, S. Identification of a novel intracellular cholesteryl ester hydrolase (carboxylesterase 3) in human macrophages: compensatory increase in its expression after carboxylesterase 1 silencing. *Am J Physiol Cell Physiol* **303**, C427-35 (2012).
 26. Sanghani, S.P. *et al.* Hydrolysis of irinotecan and its oxidative metabolites, 7-

- ethyl-10-[4-N-(5-aminopentanoic acid)-1-piperidino] carbonyloxycamptothecin and 7-ethyl-10-[4-(1-piperidino)-1-amino]-carbonyloxycamptothecin, by human carboxylesterases CES1A1, CES2, and a newly expressed carboxylesterase isoenzyme, CES3. *Drug Metab Dispos* **32**, 505-11 (2004).
27. Lian, J., Nelson, R. & Lehner, R. Carboxylesterases in lipid metabolism: from mouse to human. *Protein Cell* **9**, 178-195 (2018).
 28. Allen, S., Marsh, H. & Hodgson, A. Occurrence and conservation of the dugong (Sirenia: Dugongidae) in New South Wales. in *Proceedings of the Linnean Society of New South Wales* Vol. 125 211-216 (2004).
 29. McGowan, A.M. *et al.* Cryptic marine barriers to gene flow in a vulnerable coastal species, the dugong (*Dugong dugon*). *Marine Mammal Science* (2023).
 30. Stronen, A.V., Norman, A.J., Vander Wal, E. & Paquet, P.C. The relevance of genetic structure in ecotype designation and conservation management. *Evol Appl* **15**, 185-202 (2022).
 31. Robinson, J.A. *et al.* The critically endangered vaquita is not doomed to extinction by inbreeding depression. *Science* **376**, 635-639 (2022).
 32. Kyriazis, C.C. *et al.* Models based on best-available information support a low inbreeding load and potential for recovery in the vaquita. *Heredity (Edinb)* **130**, 183-187 (2023).
 33. Morin, P.A. *et al.* Reference genome and demographic history of the most endangered marine mammal, the vaquita. *Mol Ecol Resour* **21**, 1008-1020 (2021).

REVIEWERS' COMMENTS

Reviewer #1 (Remarks to the Author):

The authors have addressed my comments. Although I am still not entirely convinced regarding the novelty of this work, the responses were adequate.

As the authors mentioned in the rebuttal, a new study (Baker et al. 2024 – Journal of Heredity) has reported a chromosomal-level genome assembly of dugong. It is good to compare the results of Baker et al., such as the assembly and long-term N_e based on PSMC and RoH, with those observed in this study. I noticed that the heterozygosity estimated by Baker et al. was much higher than that reported in this study (~ 0.0016 Vs. 0.00088). It is better if the authors discuss the potential reasons for this discrepancy.

I see that the resequencing data has been submitted to the China National GeneBank Nucleotide Sequence Archive. I suggest that the authors provide the vcf file containing the genotypes of the 99 dugong genomes as well. This will be useful for other researchers as it will avoid weeks and months of processing fastq files to obtain the genotypes.

Reviewer #2 (Remarks to the Author):

The revised manuscript by Tian et al. presents their study on dugong genomes, offering valuable genomic resources and identifying potential genetic mechanisms underlying the Sirenian adaptation, as well as insights into dugong population demography through whole genome resequencing. Compared to the original manuscript, the revised version more clearly articulates the potential relationship between focused genetic features and marine adaptation. The authors have also provided experimental validations for the functions of these identified genetic features. While not all mechanisms are fully revealed, the revised manuscript is robust for a genomic study. Furthermore, my previous concerns have been adequately addressed, and I am largely satisfied with the revision. I do have a few minor suggestions for the authors to consider, but overall, I find the revised manuscript to be suitable for publication.

On Line 47, please include the full name of kya for clarification.

On Lines 87-88, "Hi-C" should be defined as high-throughput chromosome conformation capture.

On Lines 95-100, I noticed that Figure S1c appears after Figure S2. I would suggest reorganizing Figure S1c, perhaps merging it with Figure S3.

On Line 177, when mentioning PER2 expression, it should be referred to as a gene, so consider writing it in italic format.

On Lines 189-222, the section titled 'Gene loss and maladaptation in an altered environment' could be clearer. While I understand that adaptations can have consequences, and the loss of genes like KCNK18 may benefit marine adaptation but pose disadvantages under current temperature dynamics, this section needs more genomic, genetic, or molecular evidence to support such claims. I suggest either strengthening this part with additional evidence or reorganizing it carefully (or possibly removing it).

On Line 265, ensure consistent formatting for F_{st} as seen in Line 252.

On Lines 1429-1433, in Figure 1, gene names such as 'LCE' and 'KRT1' should also be written in italic format.

RESPONSE TO REVIEWERS' COMMENTS

We thank the reviewers for their time and effort on our manuscript. Our point-by-point responses are detailed below.

Reviewer #1 (Remarks to the Author):

Comment 1.1: *The authors have addressed my comments. Although I am still not entirely convinced regarding the novelty of this work, the responses were adequate.*

Response

We appreciate the reviewer's sentiment.

Comment 1.2: *As the authors mentioned in the rebuttal, a new study (Baker et al. 2024 – Journal of Heredity) has reported a chromosomal-level genome assembly of dugong. It is good to compare the results of Baker et al., such as the assembly and long-term N_e based on PSMC and RoH, with those observed in this study. I noticed that the heterozygosity estimated by Baker et al. was much higher than that reported in this study (~ 0.0016 Vs. 0.00088). It is better if the authors discuss the potential reasons for this discrepancy.*

Response

While Baker and colleagues¹ sequenced an individual at much higher coverage than the 99 individuals in our study ($\sim 12\times$), our dataset is at a depth suitable to estimate heterozygosity from SNPs. Moreover, the individual sequenced by Baker et al. was from Moreton Bay (MB), a population from which we re-sequenced 32 individuals. Looking at **Fig. S8c** in our manuscript, one can see that heterozygosity values vary, with MB individuals showing values (average ~ 0.0014) close to that reported by Baker and colleagues. Our ROH data on MB individuals (also obtained using ROHan) were close to that reported by Baker et al. (see **Figures S9** and **Fig. 6c**).

With regards to the long-term N_e , the mutation rate (g) employed in PSMC analysis by Baker and colleagues¹, calculated using the divergence rate between dugongs and Steller's sea cow by Le Duc and colleagues², was lower than that estimated in our study using r8s (6.25×10^{-9} vs. 2.60×10^{-8} per site per generation). Our PSMC curve on MB individuals (**Fig. 6a**) is similar to that of Baker and colleagues (their **Fig. 2C**), but the effective population size 100,000 years ago was smaller ($\sim 600,000$ vs. $\sim 12,000$ individuals). Looking at **Figure S12**, which includes three random individuals, our overall estimate still holds. We speculate that the much higher N_e from a single Moreton Bay individual in Baker and colleagues' work¹ stems from a failure to remove chromosome X before PSMC analysis – a step that can influence effective population size estimates (see^{3,4}).

In the section 'A dugong whole-genome resequencing data set' we now state: "The average heterozygosity of Moreton Bay individuals ($n=32$) mirrored an estimate from a single individual from this locality¹ (1.40×10^{-3} vs. 1.60×10^{-3})".

In the section 'Demography of *Vulnerable* and recently extinct dugongs' we now state: "Our PSMC curve of Moreton Bay individuals (**Fig. 6a** and **Figure S12**) was similar to a recent study¹ that examined a single individual from this location, but the effective population size was smaller in our dataset (e.g., $\sim 600,000$ vs. $\sim 12,000$ individuals about 100,000 years ago).

We speculate that the much higher N_e in the recent study stems from different mutation rate parameters (6.25×10^{-9} vs. 2.60×10^{-8} per site per generation in our study) or failure to remove chromosome X before PSMC analysis – a step that can influence effective population size estimates (see ^{3,4}).

Comment 1.3: *I see that the resequencing data has been submitted to the China National GeneBank Nucleotide Sequence Archive. I suggest that the authors provide the vcf file containing the genotypes of the 99 dugong genomes as well. This will be useful for other researchers as it will avoid weeks and months of processing fastq files to obtain the genotypes.*

Response

We agree that providing the VCF file would be very valuable, and we have uploaded the data to the European Nucleotide Archive (accession number PRJNA1114306).

Reviewer #2 (Remarks to the Author):

Comment 2.1: *The revised manuscript by Tian et al. presents their study on dugong genomes, offering valuable genomic resources and identifying potential genetic mechanisms underlying the Sirenian adaptation, as well as insights into dugong population demography through whole genome resequencing. Compared to the original manuscript, the revised version more clearly articulates the potential relationship between focused genetic features and marine adaptation. The authors have also provided experimental validations for the functions of these identified genetic features. While not all mechanisms are fully revealed, the revised manuscript is robust for a genomic study. Furthermore, my previous concerns have been adequately addressed, and I am largely satisfied with the revision. I do have a few minor suggestions for the authors to consider, but overall, I find the revised manuscript to be suitable for publication.*

Response

We greatly appreciate that reviewer's response to our revision.

Comment 2.2: *On Line 47, please include the full name of kya for clarification.*

Response

The sentence now reads '10.7 thousand years ago'.

Comment 2.3: *On Lines 87-88, "Hi-C" should be defined as high-throughput chromosome conformation capture.*

Response

Fixed.

Comment 2.4: *On Lines 95-100, I noticed that Figure S1c appears after Figure S2. I would suggest reorganizing Figure S1c, perhaps merging it with Figure S3.*

Response

Done.

Comment 2.5: *On Line 177, when mentioning PER2 expression, it should be referred to as a gene, so consider writing it in italic format.*

Response

Agree.

Comment 2.6: On Lines 189-222, the section titled 'Gene loss and maladaptation in an altered environment' could be clearer. While I understand that adaptations can have consequences, and the loss of genes like *KCNK18* may benefit marine adaptation but pose disadvantages under current temperature dynamics, this section needs more genomic, genetic, or molecular evidence to support such claims. I suggest either strengthening this part with additional evidence or reorganizing it carefully (or possibly removing it).

Response

We have now attempted to more clearly show that our proposed functional effect of *KNCK18* loss requires additional evidence.

We now state “Although speculative, we hypothesize that loss of *KCNK18* decreases sirenian temperature tolerance (**Figure 4e**) and that CSS is similar to semelparity in marsupials^{5,6} in that a progressive and systemic deterioration of body condition and physiological function is mediated by an endocrine factor, perhaps from elevated levels of the stress hormone cortisol”

Comment 2.7: On Line 265, ensure consistent formatting for *Fst* as seen in Line 252.

Response

Fixed (changed all to *F_{ST}*).

Comment 2.8: On Lines 1429-1433, in Figure 1, gene names such as '*LCE*' and '*KRT1*' should also be written in italic format.

Response

LCE refers to the late cornified envelope gene family, so it should not be in italics. *KRT1* here should be ‘type I keratins’, not *KRT1* – we apologize for this error and have updated the figure. We have updated the figure and figure legend to better reflect this.

References

- 1 Baker, D. N. *et al.* A chromosome-level genome assembly for the dugong (Dugong dugon). *J Hered*, doi:10.1093/jhered/esae003 (2024).
- 2 Le Duc, D. *et al.* Genomic basis for skin phenotype and cold adaptation in the extinct Steller's sea cow. *Sci Adv* **8**, eabl6496, doi:10.1126/sciadv.abl6496 (2022).
- 3 Li, H. & Durbin, R. Inference of human population history from individual whole-genome sequences. *Nature* **475**, 493-496, doi:10.1038/nature10231 (2011).
- 4 Cousins, T., Tabin, D., Patterson, N., Reich, D. & Durvasula, A. Accurate inference of population history in the presence of background selection. *bioRxiv*, doi:10.1101/2024.01.18.576291 (2024).
- 5 Tian, R. *et al.* A chromosome-level genome of *Antechinus flavipes* provides a reference for an Australian marsupial genus with male death after mating. *Mol Ecol Resour* **22**, 740-754, doi:10.1111/1755-0998.13501 (2022).
- 6 Naylor, R., Richardson, S. J. & McAllan, B. M. Boom and bust: a review of the physiology of the marsupial genus *Antechinus*. *J Comp Physiol B* **178**, 545-562, doi:10.1007/s00360-007-0250-8 (2008).